# The role of interspecies recombination in the evolution of antibiotic-resistant pneumococci

Joshua C D'Aeth[1]*, Mark PG van der Linden[2], Lesley McGee[3], Herminia de Lencastre[4,5], Paul Turner[6,7], Jae-Hoon Song[8], Stephanie W Lo[9], Rebecca A Gladstone[9], Raquel Sá-Leão[10], Kwan Soo Ko[8], William P Hanage[11], Robert F Breiman[12], Bernard Beall[3], Stephen D Bentley[9], Nicholas J Croucher[1]*, The GPS Consortium

[1]MRC Centre for Global Infectious Disease Analysis, Department of Infectious Disease Epidemiology, Imperial College London, London, United Kingdom; [2]Institute for Medical Microbiology, National Reference Center for Streptococci, University Hospital RWTH Aachen, Aachen, Germany; [3]Respiratory Diseases Branch, Centers for Disease Control and Prevention, Atlanta, United States; [4]Laboratory of Molecular Genetics, Instituto de Tecnologia Química e Biológica, Universidade Nova de Lisboa, Oeiras, Portugal; [5]Laboratory of Microbiology and Infectious Diseases, The Rockefeller University, New York, United States; [6]Cambodia Oxford Medical Research Unit, Angkor Hospital for Children, Siem Reap, Cambodia; [7]Centre for Tropical Medicine and Global Health, Nuffield Department of Medicine, University of Oxford, Oxford, United Kingdom; [8]Department of Molecular Cell Biology, Sungkyunkwan University School of Medicine, Suwon, Republic of Korea; [9]Parasites & Microbes, Wellcome Sanger Institute, Wellcome Genome Campus, Hinxton, United Kingdom; [10]Laboratory of Molecular Microbiology of Human Pathogens, Instituto de Tecnologia Química e Biológica, Universidade Nova de Lisboa, Oeiras, Portugal; [11]Center for Communicable Disease Dynamics, Harvard T.H. Chan School of Public Health, Boston, United States; [12]Department of Global Health, Rollins School of Public Health, Emory University, Atlanta, United States

*For correspondence:
j.daeth17@imperial.ac.uk (JCD'A);
n.croucher@imperial.ac.uk (NJC)

Group author details:
The GPS Consortium See page 27

**Abstract** Multidrug-resistant *Streptococcus pneumoniae* emerge through the modification of core genome loci by interspecies homologous recombinations, and acquisition of gene cassettes. Both occurred in the otherwise contrasting histories of the antibiotic-resistant *S. pneumoniae* lineages PMEN3 and PMEN9. A single PMEN3 clade spread globally, evading vaccine-induced immunity through frequent serotype switching, whereas locally circulating PMEN9 clades independently gained resistance. Both lineages repeatedly integrated Tn*916*-type and Tn*1207.1*-type elements, conferring tetracycline and macrolide resistance, respectively, through homologous recombination importing sequences originating in other species. A species-wide dataset found over 100 instances of such interspecific acquisitions of resistance cassettes and flanking homologous arms. Phylodynamic analysis of the most commonly sampled Tn*1207.1*-type insertion in PMEN9, originating from a commensal and disrupting a competence gene, suggested its expansion across Germany was driven by a high ratio of macrolide-to-β-lactam consumption. Hence, selection from antibiotic consumption was sufficient for these atypically large recombinations to overcome species boundaries across the pneumococcal chromosome.

## Introduction

Infections caused by *Streptococcus pneumoniae* (the pneumococcus) remain a leading cause of death worldwide in children under the age of five (*Wahl et al., 2018*; *GBD 2015 Mortality and Causes of Death Collaborators, 2016*). This nasopharyngeal commensal and respiratory pathogen causes a range of severe infections in both infants and adults, including pneumonia, sepsis, and meningitis. These have a high mortality rate, which is further increased when the causative pneumococcus is resistant to antibiotics (*Cassini et al., 2019*; *Feikin et al., 2000*). This presents a worrying challenge to clinicians, with treatment options decreasing for resistant infections (*Roca et al., 2015*). As the pneumococcus is endemic worldwide, its ability to develop antibiotic resistance is a global challenge (*Appelbaum, 1987*). High levels of resistance have been observed in Africa (*Chaguza et al., 2017*), Asia, and the Americas. Even in Europe, where resistance is less common, deaths attributable to penicillin-resistant pneumococci have been rising over the past 15 years (*Cassini et al., 2019*). Furthermore, this resistance is under selection by community antibiotic consumption, a substantial proportion of which is often attributable to common noninvasive pneumococcal diseases , such as otitis media (*Dewé et al., 2019*; *Vergison et al., 2010*).

There are two main mechanisms by which pneumococci gain antibiotic resistance: the modification of core genes encoding antibiotic targets, often through homologous recombination, and the acquisition of specialized resistance genes on mobile genetic elements (MGEs) (*Dewé et al., 2019*; *Croucher et al., 2014a*). As the plasmid repertoire of *S. pneumoniae* is limited to two types of cryptic elements (*Smith and Guild, 1979*; *Romero et al., 2007*; *Schuster et al., 1998*), the MGEs that contribute most to the spread of antibiotic resistance are integrative and conjugative elements (ICEs) (*Croucher et al., 2009*; *Croucher et al., 2014b*). These MGEs contain integrase genes that mediate their insertion within the host cell genome (*Johnson and Grossman, 2015*) and, given their resistance gene cargo, have been referred to as 'king makers' of bacterial lineages (*Baker et al., 2018*). ICEs mobilize between cells through conjugation, a highly efficient method of DNA transfer, involving a pilus formed between donor and recipient cells protecting transferred DNA from the external environment (*Cabezón et al., 2015*). This has enabled conjugation to transfer elements across a broad range of bacterial taxa (*von Wintersdorff et al., 2016*; *Roberts and Mullany, 2009*; *Musovic et al., 2006*), which seems the most likely explanation as to how antibiotic resistance genes originally entered the *S. pneumoniae* population (*Croucher et al., 2009*).

The most important ICEs driving the spread of antibiotic resistance in pneumococci are related to Tn*916*, the first ICE to be discovered (*Wozniak and Waldor, 2010*; *Roberts and Mullany, 2011*). This element confers tetracycline resistance via the *tetM* gene and forms composite elements that can confer resistance to macrolides, aminoglycosides, streptogramins, and lincosamides through the integration of sequences such as the Mega cassette, Omega cassette, and Tn*917* elements (*Croucher et al., 2011*). Tn*916*-type elements are found in the majority of antibiotic-resistant bacterial pathogens which are considered a priority by the WHO (*Roberts and Mullany, 2009*; *Roberts and Mullany, 2011*; *WHO, 2017*). However, the distribution of Tn*916*-type elements in *S. pneumoniae* is a particular puzzle as in vitro studies have shown that they appear unable to conjugate between pneumococci, although pneumococci themselves can be donors to other streptococci (*Mingoia et al., 2011*; *Cochetti et al., 2007*). The short cassettes that integrate into Tn*916* also lack their own self-mobilization machinery, which is even true for the longer form of the Mega cassette, Tn*1207.1*. Hence, the contribution of conjugation to the spread of these ICE is not clear.

An alternative mechanism by which Tn*916* might spread is transformation, which describes the uptake of extracellular DNA into cells that have reached a competent state (*Johnston et al., 2014*). Originally discovered in the pneumococcus, natural transformation is tightly controlled by the host cell, which encodes all the required machinery (*Johnston et al., 2014*). Once DNA has been imported into the cell, it can then be integrated into the chromosome via homologous recombination. However, there are two important limitations on the dissemination of MGEs between species through transformation.

The first limitation is the inhibition of homologous recombination by sequence divergence, which reduces the integration of sequence from other species into host chromosomal DNA (*Croucher et al., 2016*; *Kung et al., 2013*; *Mostowy et al., 2017a*; *Croucher et al., 2012*). The decline is exponential as sequence divergence between the donor and recipient increases (*Majewski et al., 2000*). This is thought to result from the minimum effective processing segment

(MEPS), the shortest length of continuous sequence identity required for efficient recombination, estimated to be 27 bp for pneumococci (*Croucher et al., 2012*). Yet, this barrier was not sufficient to prevent interspecies recombinations facilitating the emergence of β-lactam-resistant pneumococci. This involved the formation of 'mosaic' versions of multiple genes encoding targets of β-lactam antibiotics (most commonly, *pbp1a*, *pbp2b*, and *pbp2x*) that were a mixture of sequence from *S. pneumoniae* and the related oronasopharyngeal commensal streptococcal species, *Streptococcus mitis* and *Streptococcus oralis* (*von Wintersdorff et al., 2016*; *Dowson et al., 1993*; *Dowson et al., 1990*). The mosaicism reflected the imported fragments being much smaller than a typical gene, although there was also evidence of these recombinations causing diversification in the flanking regions of the chromosome (*Enright and Spratt, 1999*).

The second limitation is that transformation requires the importation of both the intact locus and two flanking 'homologous arms,' in which recombination crossovers can occur. Both arms must match the host chromosome, and no cleavage of the imported DNA must occur between them during its uptake into the cell. Hence, the efficiency of uptake declines with the length of the inserted locus between the two arms (*Apagyi et al., 2018*). Therefore, while in vitro studies have shown that interspecies transfer of MGEs via transformation is possible (*Domingues et al., 2012*), these transfers were of a low frequency and only spread shorter resistance cassettes. As such, transformation has been primarily considered a mechanism facilitating the intraspecific spread of small MGEs (*Chancey et al., 2015*).

Ergo, although there are multiple mechanisms by which MGEs may be acquired, their relative contributions are not known. This is particularly challenging when antibiotic resistance loci are ubiquitous throughout a strain, as Tn*916*-type elements are within many antibiotic-resistant *S. pneumoniae* lineages (*Croucher et al., 2014a*; *Croucher et al., 2011*; *Croucher et al., 2014c*), making the process underlying the MGE's acquisition difficult to infer. Therefore, we investigated two globally distributed pneumococcal lineages in which antibiotic resistance MGEs are common, but not conserved.

The first is the Spain[9V]-3, or PMEN3 lineage, which is within strain GPSC6 (clonal complex 156 by multilocus sequence typing [MLST]) (*Gladstone et al., 2019*). PMEN3 was first documented in Spain in 1988 with a serotype 9V capsule. It was later detected in France, the USA, and South America (*Coffey et al., 1991*; *Lefèvre et al., 1995*; *Corso et al., 1998*; *Tomasz et al., 1998*). By 2000, 55% of all penicillin-resistant disease isolates in South America were from the PMEN3 lineage (*Tomasz et al., 1998*).

The second is the England[14]-9, or PMEN9 lineage, which is within strain GPSC18 (clonal complex 9 or 15 by MLST). This was first described in the UK in 1996, and it has been isolated across Europe, the Americas, and Asia (*Gherardi et al., 2000*; *Sá-Leão et al., 2000*; *Tsolia et al., 2002*). PMEN9 was the most common lineage causing penicillin-resistant invasive pneumococcal disease (IPD) in the USA, and the most common lineage causing macrolide-resistant IPD in Germany, just prior to the introduction of infant immunization with the heptavalent polysaccharide conjugate vaccine (PCV7) (*Kim et al., 2016*; *Bley et al., 2011*; *Imöhl et al., 2010*).

Both lineages exhibit variability in their resistance to tetracyclines and macrolides across countries, suggesting frequent acquisition or loss of elements related to Tn*916* and Tn*1207.1* (*Zemlicková et al., 2007*; *Cochetti et al., 2005*; *Càmara et al., 2018*; *Corso et al., 2009*). Therefore, we analyzed the distribution of antibiotic resistance loci within these lineages, and then expanded the study across the species with the wider Global Pneumococcal Sequencing (GPS) project data (*Gladstone et al., 2019*; *GPS, 2021*). Using a mixture of genomic approaches, we assessed the distribution of antibiotic resistance loci, determined the mechanisms by which they were imported into these lineages, and characterized how these genotypes have adapted to local antibiotic use as they have spread globally.

## Results

### Divergent genomic epidemiology of antibiotic-resistant pneumococci

The PMEN3 and PMEN9 lineages had contrasting evolutionary and transmission histories. The phylogeny representing the evolution of PMEN3 was constructed from isolates of GPSC6 (treated as synonymous with PMEN3 here on in), which were collected from 31 countries over 23 years (1992–

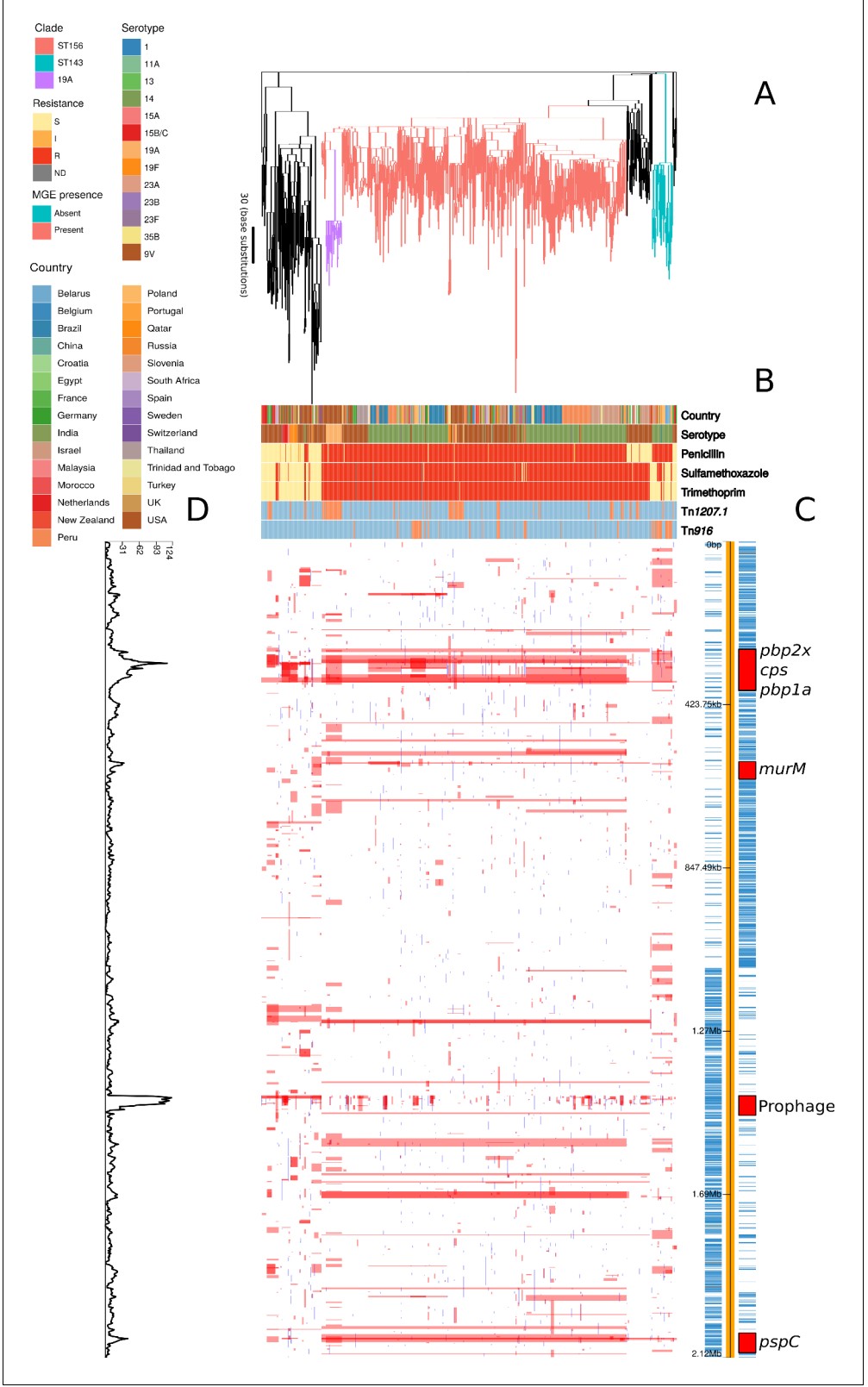

**Figure 1.** Phylogenomic analysis of PMEN3 lineage. (**A**) Maximum likelihood phylogeny generated from the nonrecombinant regions of the whole-genome alignment of 669 isolates from the PMEN3 lineage. Branches are colored by clade, as identified in the key. Units for the scale bar are the number of point mutations along a branch. (**B**) Bars highlighting the country of origin; serotype; resistance to penicillin, trimethoprim and

*Figure 1 continued on next page*

*Figure 1 continued*

sulfamethoxazole, and the presence of the mobile genetic elements (MGEs) Tn*1207.1* and Tn*916* among isolates. The abbreviated resistance phenotypes are resistant (R), intermediate-level resistant (I), and susceptible (S). Bars map across to isolates on the phylogeny. (C) Simplified genome annotation of the PMEN3 reference isolate RMV4. The highlighted regions correspond to peaks of recombination event frequency. Blue bars represent individual genes annotated within the assembly. (D) Distribution of recombination events across the PMEN3 lineage. In the upper half of the graph, red bars indicate recombination events occurring on internal nodes in the tree, which were subsequently inherited by multiple descendant isolates. These bars align with isolates in the phylogeny in section A and map to regions in the genome annotated in section C. Blue bars indicate recombination events on terminal branches of the tree, identified in only one isolate. In the bottom half of the graph, the line represents the frequency of recombination events along the genome's length.

The online version of this article includes the following source data and figure supplement(s) for figure 1:

**Source data 1.** Sequence and epidemiological data for *S. pneumoniae* samples.
**Figure supplement 1.** Root-to-tip analysis of PMEN3 lineage.
**Figure supplement 2.** Serotype switching events across the PMEN3 lineage.

---

2015; *Figure 1*). This range was sufficient for the estimation of a molecular clock (*Figure 1—figure supplement 1*). This estimated the most recent common ancestor (MRCA) existed in 1949 (95% credible interval of 1930–1962) and the lineage had a molecular clock rate of $1.71 \times 10^{-6}$ substitutions per site per year (95% credible interval of $1.56 \times 10^{-6}$ to $1.87 \times 10^{-6}$ substitutions per site per year). The PMEN3 phylogeny was dominated by the 491-isolate ST156 clade, which was found in 27 countries mainly from South America (192 isolates), North America (90 isolates) and Europe (85 isolates). There was also a smaller, 33 isolate clade of ST143 isolates, which was found in Poland (11 isolates), Belarus (8 isolates), and six other countries. Most of the PMEN3 isolates were either of the ancestral serotype 9V or serotype 14, with changes between these two serotypes accounting for 9 of 36 serotype switches reconstructed within the clade (*Figure 1—figure supplement 2*). Both serotypes 9V and 14 were targeted by the PCV7 vaccine. However, a clade of 26 ST156 isolates from the USA of serotype 19A, not included in PCV7, were derived from a MRCA estimated to exist in 2000 (95% credible interval of 1999–2001). This coincides with the date of PCV7's introduction into the USA, consistent with these switched isolates evading the vaccine and persisting until the 13-valent conjugate vaccine (PCV13), which includes 19A, was introduced (*Kim et al., 2016*). In total 13 serotypes were found in the PMEN3 lineage, of which 7 (11A, 13, 15A, 15B/C, 23A, 23B, and 35B) are not found in the PCV13 vaccine. Hence, a single PMEN3 clade has rapidly diversified its surface antigens as it has frequently disseminated between countries.

By contrast, the phylogeny representing the evolution of the PMEN9 lineage, constructed from isolates of the GPSC18 strain (treated as synonymous with PMEN9 from here on in), was split into multiple clades separated by deep branches (*Figure 2*). Even when excluding the outlying serotype 7C isolates, the only discernible molecular clock signal suggested this strain was centuries old (*Figure 2—figure supplement 1*). Despite this age, the individual clades were generally regionally confined. The largest clade was associated with Germany (accounting for 166 of the 250 isolates), with other representatives from Slovenia and China. Other clades were associated with the USA (accounting for 91 of the 98 isolates), South Africa (accounting for 68 of the 73 isolates), and China (accounting for 18 of 45 isolates). All the isolates in the three largest clades expressed serotype 14, as did 93% of all isolates in this phylogeny. Only nine serotype switches were identified across PMEN9, including switches to 19F and 23F in the Chinese clade (*Figure 2—figure supplement 2*). In total, there were six serotypes present within the collection, only two of which (16F and 7C) were not found in the PCV13 vaccine. Overall, there was little evidence of frequent intercontinental transmission or antigenic diversification with this set of isolates. Hence, genomics suggests differing histories for these lineages, despite them both being internationally disseminated antibiotic-resistant *S. pneumoniae*, commonly expressing the invasive serotype 14 and having identical sampling approaches.

## Variation in transformation rates and imported sequence properties

The two lineages also differed in the patterns of recombination across their genomes. In the PMEN3 reference genome, there is a high density of recombinations around a 45 kb prophage region, indicating frequent infection by phage. Exclusion of these recombination events allowed estimation of

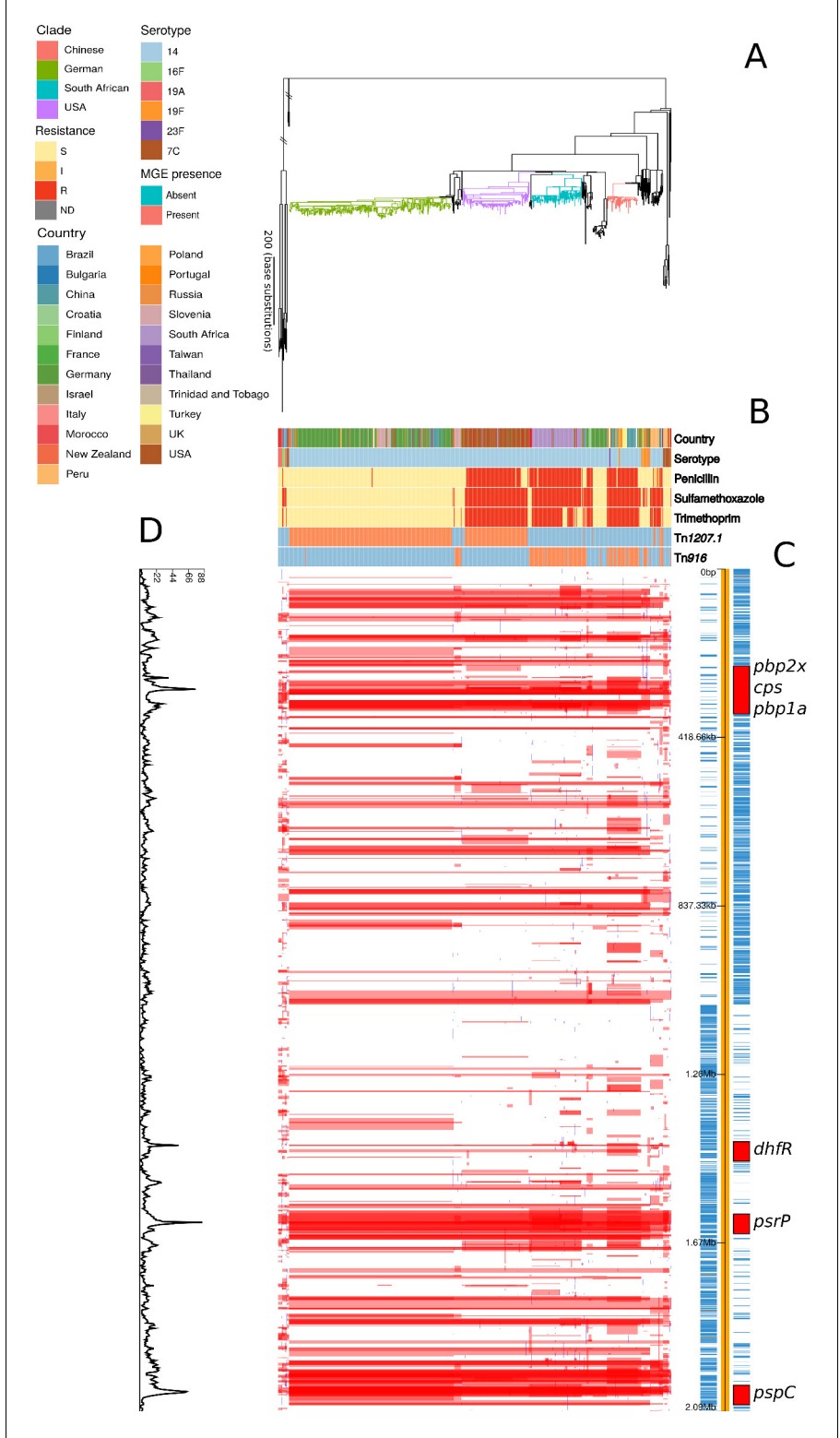

**Figure 2.** Phylogenomic analysis of PMEN9 lineage. (**A**) Maximum likelihood phylogeny generated from the nonrecombinant regions of a whole-genome alignment of 575 isolates from the PMEN9 lineage. Branches are colored by clade, as identified in the key. Units for the scale bar are the number of point mutations along a branch. (**B**) Bars highlighting the country of origin; serotype; resistance to penicillin, trimethoprim, and sulfamethoxazole, and the presence of the mobile genetic elements (MGEs) Tn*1207.1* and Tn*916* among isolates.

*Figure 2 continued on next page*

*Figure 2 continued*

The abbreviated resistance phenotypes are resistant (R), intermediate-level resistant (I), and susceptible (S). Bars map across to isolates on the phylogeny. (**C**) Simplified annotated genome of the PMEN9 reference isolate INV200. The highlighted regions correspond to peaks of recombination event frequency. Blue bars represent individual genes annotated within the assembly. (**D**) Distribution of recombination events across the PMEN9 lineage. In the upper half of the graph, red bars indicate recombination events occurring on internal nodes in the tree, which were subsequently inherited by multiple descendant isolates. These bars are aligned with isolates in the phylogeny in section A and map to regions in the genome annotated in section C. Blue bars indicate recombination events on terminal nodes of the tree, occurring in only one isolate. In the bottom half of the graph, the line represents the frequency of recombination events along the genome's length.

The online version of this article includes the following figure supplement(s) for figure 2:

**Figure supplement 1.** Root-to-tip analysis of PMEN9 lineage.

**Figure supplement 2.** Serotype switching events across the PMEN9 lineage.

the overall ratios of base substitutions resulting from homologous recombination relative to point mutations (*r/m*). Consistent with its more rapid serological diversification, *r/m* was higher in PMEN3 (13.1) than PMEN9 (7.7).

This difference in *r/m* could be due to the two lineages differing in three ways: (i) in the number of recombinations, (ii) in the length of recombination events, or (iii) in the sources of their recombination events, with more divergent sources increasing the *r/m*.

The first explanation partially accounted for the difference: there were 0.115 recombinations per point mutation in the PMEN3 reconstruction compared to 0.093 per points mutation in PMEN9. Comparing the properties of the recombination events revealed no substantial difference in their length distribution (*Figure 3*). However, PMEN3 generally imported sequences with a significantly higher SNP density, with a median SNP density of 11.8 SNPs/kb of sequence imported, compared to PMEN9, which had a median SNP density of 9.2 SNPs/kb (Mann–Whitney U = 4,162,888, n1 = 2613, n2 = 2823, two-sided, p<$2.2 \times 10^{-16}$). Therefore, the difference in *r/m* between the two lineages reflected both the increased frequency of recombination in the PMEN3 lineage, and the increased diversity of the imported sequence.

Several peaks of recombination within the chromosome corresponded to loci likely to be under immune selection. In PMEN9, there was an elevated density of recombinations affecting the *psrP* gene, encoding the antigenic pneumococcal serine-rich repeat surface protein. Additionally, the antigenic Pneumococcal Surface Protein C, encoded by *pspC*, is a recombination hotspot in both lineages. Both these genes are highly diverse in pneumococcal populations, and encode proteins eliciting strong immune responses from hosts (*Croucher et al., 2017*).

Both lineages also had large recombination hotspots at their *cps* loci, which determine an isolate's serotype (*Mostowy et al., 2017a*; *Salter et al., 2012*). These loci underwent frequent diversification in the serologically diverse PMEN3, with fewer events at this locus in PMEN9 (*Figure 1—figure supplement 2* and *Figure 2—figure supplement 2*). Within PMEN3, for switches from 9V to 14, the median recombination block size spanning the 20 kb *cps* locus was 26.5 kb in length. These recombination blocks also frequently encompassed the neighbouring *pbp1a* and *pbp2x* genes, encoding penicillin-binding proteins (PBPs) involved in penicillin resistance. In PMEN9, three of the seven recombination events causing serotype switches affected either *pbp1a* or *pbp2x*; this proportion increased to over 75% (26 of 34) of the recombinations associated with a serotype switch in PMEN3.

## Emergence of β-lactam resistance

Penicillin resistance was predicted using a random forest (RF) model trained on the PBP transpeptidase domains (TPDs; see Materials and methods), which categorized isolates using the pre-2008 CLSI meningitis resistance breakpoints (*Figure 4—figure supplement 1*). Using this classification of the PMEN9 collection, 61% of isolates were susceptible to penicillin (recorded or predicted minimum inhibitory concentration (MIC) ≤ 0.06 µg/ml), 38% of isolates were resistant (MIC ≥ 0.12 µg/ml) and the remaining 1% were classified as intermediately resistant (0.06 µg/ml < MIC < 0.12 µg/ml; *Figure 2*). However, in the PMEN3 collection only 20% of isolates were susceptible to penicillin, with 79% being resistant and the remaining 1% classified as intermediately resistant (*Figure 1*).

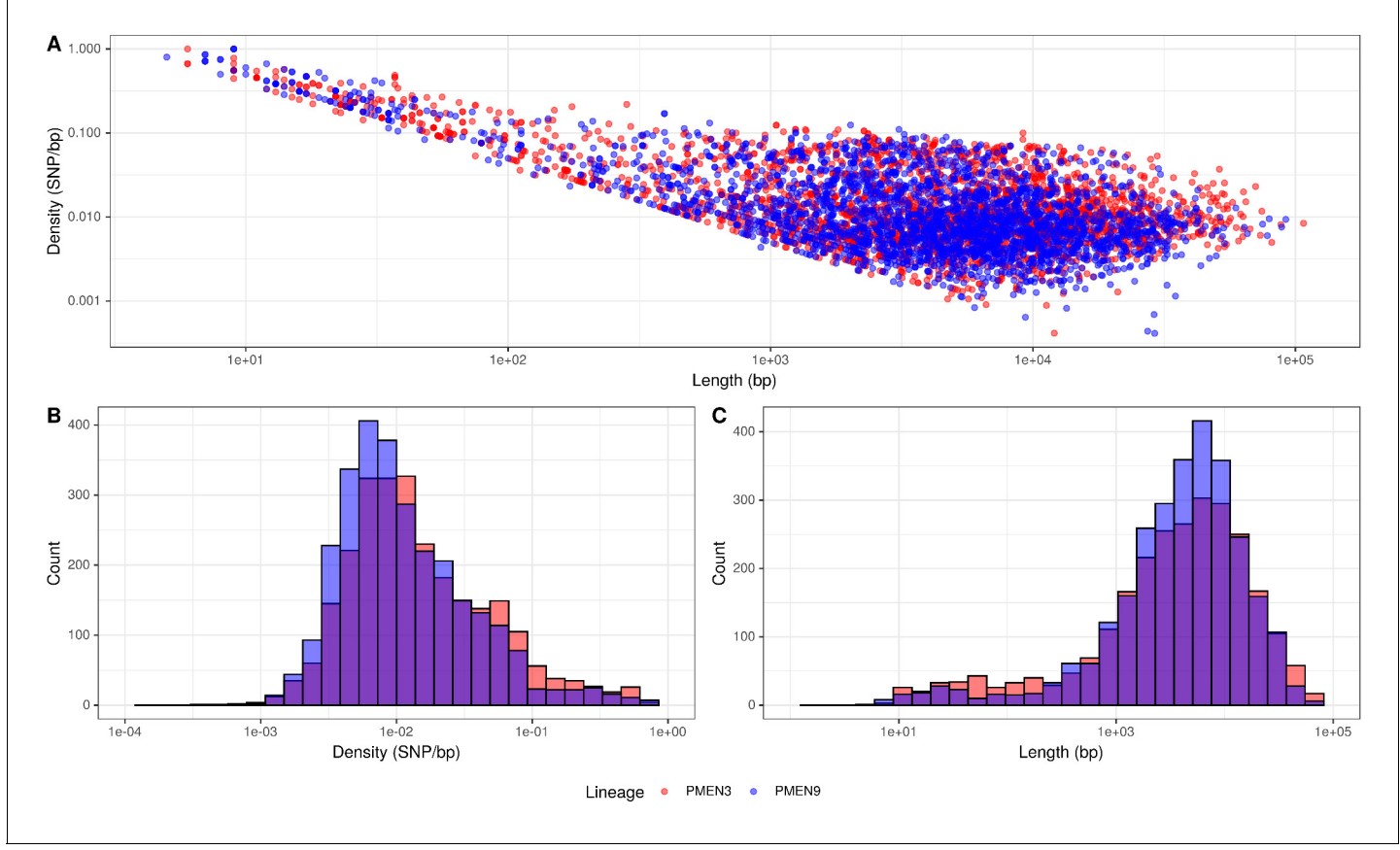

**Figure 3.** Summary of recombination differences between the PMEN3 and PMEN9 lineages. (**A**) Plotting the SNP density of recombination events relative to their length across both lineages. Dots represent individual recombination events and are colored by the lineage in which they were inferred. (**B**) Overlaid histograms of the SNP densities of recombination events across PMEN3 and PMEN9. (**C**) Overlaid histograms of the length, in bases, of each recombination event across PMEN3 and PMEN9.

Across the two PMEN lineages, there were 35 reconstructed changes in resistance profile for penicillin. The most common alteration was acquisition of resistance by sensitive isolates, with 16 instances in the two lineages (46% of events). There were also seven instances of resistant isolates reverting to penicillin sensitivity across the collections. In 20 of the 35 alterations in resistance profile, the evolutionary reconstruction identified at least one of the three resistance-associated *pbp* genes was altered by a concomitant recombination event.

In PMEN3, 99% of the ST156 clade was penicillin resistant. Recombinations altered *pbp1a*, *pbp2b*, and *pbp2x* at the base of this clade (*Figure 1*). Combining the time-calibrated phylogeny with the ancestral state reconstruction of penicillin resistance showed the penicillin-resistant proportion of GPSC6 increased throughout the early 1980s, driven by the expansion of the ST156 clade, which originated around 1984 (95% credible interval 1982–1986) (*Figure 4*). This expansion of resistant lineages continued until the early 2000s when it then plateaued within the strain from roughly 2010 onward.

The highest MICs within the ST156 clade (up to 8 μg/ml) were associated with the vaccine escape serotype 19A clade of isolates from the USA (*Figure 4—figure supplement 2*). This was a consequence of a 53 kb recombination spanning the *cps* locus, which caused the alteration in serotype, also spanning *pbp1a* and *pbp2x* (*Figure 1—figure supplement 2*). Hence, the PCV7-escape recombination also reduced susceptibility to antibiotics. The converse situation was observed for a single ST156 clade member that had reverted to susceptibility. A 53 kb recombination event, causing a switch from serotype 9V to 15B/C (*Figure 1—figure supplement 2*), restored the ancestral, susceptible versions of *pbp2x* and *pbp1a*.

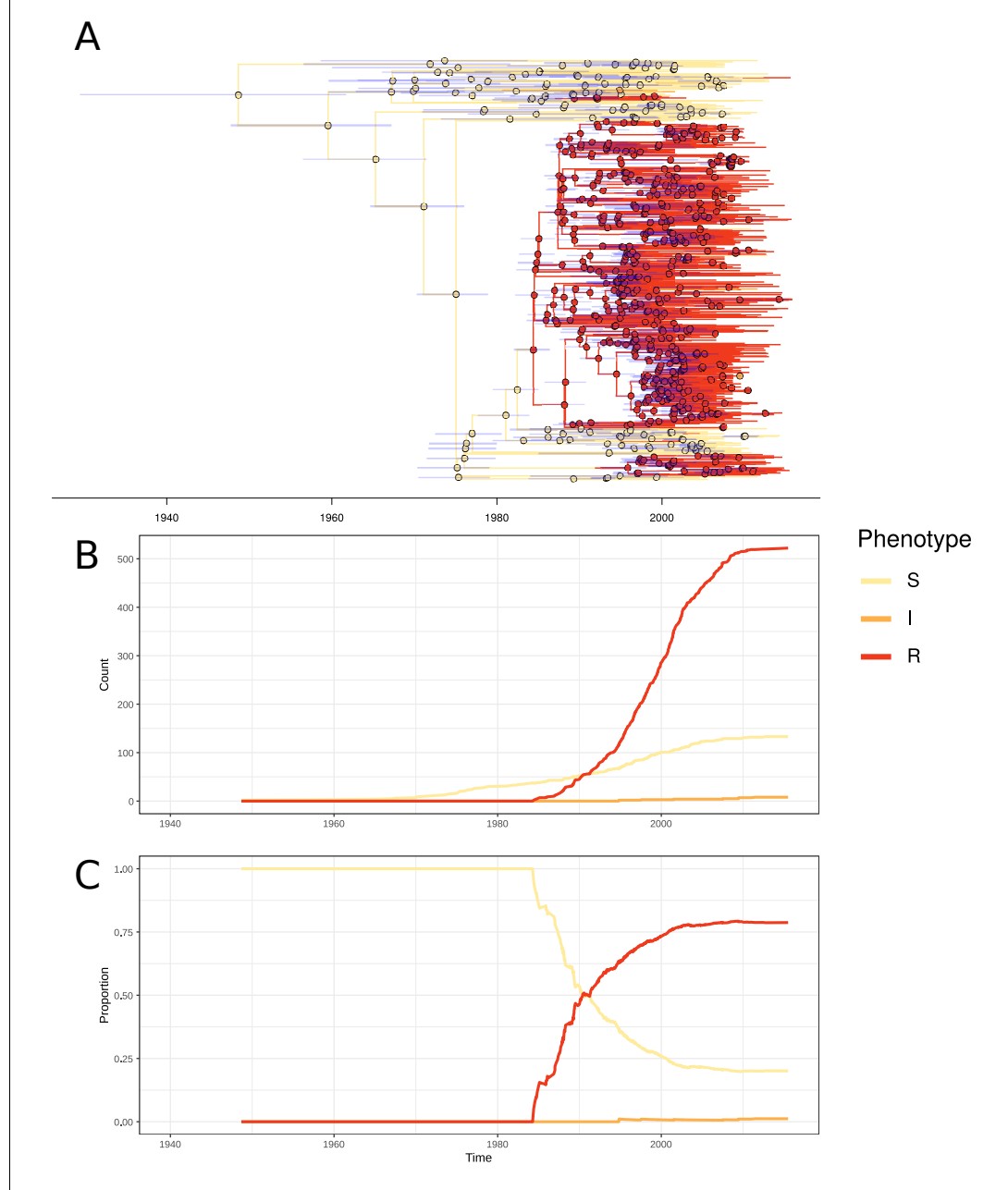

**Figure 4.** Emergence of resistant lineages within PMEN3 through time. (A) Time-calibrated phylogeny of PMEN3. Branches are colored by inferred resistance phenotype. Pie charts present at nodes represent the inferred probability of each phenotype by an ancestral reconstruction. Blue bars across the nodes represent the 95% credible interval for the age of the node. (B) The reconstructed absolute number of branches per resistant phenotype through time. (C) The proportion of total branches over time reconstructed as having each of the resistance phenotypes.

The online version of this article includes the following figure supplement(s) for figure 4:

**Figure supplement 1.** Comparisons of category agreement across different penicillin resistance breakpoints.

**Figure supplement 2.** Histograms of recorded minimum inhibitory concentration (MIC) values for penicillin across the PMEN3 and PMEN9 lineages.

**Figure supplement 3.** Origin of *pbp* genes for penicillin-resistant isolates.

**Figure supplement 4.** Analysis of the origin of the *murM* gene across the PMEN3 and PMEN9 lineages.

In contrast to PMEN3, penicillin resistance emerged independently in different locations within PMEN9. The USA and South African clades appear to have both separately gained resistance in a stepwise manner. At the base of the highly resistant USA clade, there was a 3.2 kb recombination spanning the *pbp2x* gene (*Figure 4—figure supplement 2*). Subsequent recombinations modifying the *pbp1a*, then *pbp2b*, genes further increased penicillin resistance. Similarly, within the South African clade, *pbp2b* and *pbp2x* were both modified by recombination in the isolates' MRCA. Resistance was elevated in a subset of isolates through further modification of *pbp1a* through recombination. Alteration in the *pbp2x* and *pbp2b* genes are the first steps towards resistance, with *pbp1a* modifications required for higher levels of resistance, although isolates with solely a mosaic *pbp2x* gene have been found to be resistant to penicillin (*Zapun et al., 2008*). In general, we observed penicillin resistance rapidly emerged and spread worldwide in PMEN3, whereas PMEN9 exhibits repeated, stepwise acquisition of modified *pbp* genes in multiple regions.

## Role of interspecies transformation in β-lactam resistance

As penicillin resistance was originally demonstrated to involve the acquisition of sequence from related commensal streptococci (*Dowson et al., 1993*; *Dowson et al., 1990*; *Laible et al., 1991*), the origin of these *pbp* genes within recombination events was analyzed with a simple statistic, $\gamma$ (see Materials and methods). This had a value of 1 if a recombination was likely to originate within *S. pneumoniae*, else was lower if it came from a donor of a related species. For gains of resistance from sensitivity across the PMEN3 and PMEN9 lineages, the median $\gamma$ score for *pbp1a* was 1.0, while for *pbp2b* it was 0.95 and for *pbp2x* it was 0.72. This pattern was exemplified by the emergence of ST156 (*pbp1a* = 1.0, *pbp2b* = 0.92 *pbp2x* = 0.62), suggesting that the *pbp2x* and *pbp2b* loci were most affected by recombination with nonpneumococcal streptococci. By contrast, the $\gamma$ score for the *pbp1a* and *pbp2x* genes was 1.0 for the reversion to penicillin susceptibility within ST156, consistent with a restoration of the ancestral pneumococcal alleles (*pbp2b* was not present within a recombination block for this alteration).

The origin of resistance-associated *pbp* alleles was analyzed across the species using the GPS collection. Penicillin resistance levels across 621 GPSCs were estimated using the RF method, as described for the PMEN3 and PMEN9 lineages (see Materials and methods). Overall, the RF method generated penicillin resistance phenotype predictions for 19,962 of 20,043 (99%) isolates. Most isolates (64%) were susceptible to penicillin, while 30% were classified as resistant, and the remaining 6% as intermediately resistant. Ancestral state reconstructions identified 338 changes in penicillin resistance phenotypes across the 17,590 isolates present within 146 resistance-associated GPSCs (see Materials and methods). The joint most common changes were susceptible to resistant, and susceptible to intermediately resistant, occurring 117 times (35%) each. In total, 184 of the 338 alterations (54%) were associated with an inferred recombination event affecting at least one of *pbp1a*, *pbp2b,* or *pbp2x*. The *pbp2x* gene was most frequently identified as being altered by recombination, occurring in 100 of the 184 alterations in nonsusceptibility associated with a recombination.

As was the case with the PMEN lineages, the emergence of resistance from susceptible genotypes was often associated with parts of the *pbp2x* and *pbp2b* genes being imported from other species (indicated by $\gamma < 1$; *Figure 4—figure supplement 3*). The median $\gamma$ score for *pbp2b* was 0.96, and the gene had a $\gamma$ score below 1 in 22 gains of resistance. While the median $\gamma$ score for *pbp2x* was 1.0, there were 15 gains of resistance where $\gamma$ was below 1. The median $\gamma$ score of *pbp1a* was 1.0, with only one instance where its score was below 1, again consistent with little modification by interspecies exchanges. However, where resistant isolates reverted to susceptibility, across all three genes the median $\gamma$ score was 1.0, indicating within-species recombinations could cause the loss of resistance.

## Evolution of resistance through recombination at other core loci

In PMEN3, there were further peaks in recombination frequency around the *murM* gene (*Figure 1*), which encodes an enzyme involved in cell wall biosynthesis (*Filipe et al., 2000*) that has also been implicated in affecting penicillin resistance (*Dewé et al., 2019*; *Filipe and Tomasz, 2000*). Yet compared to the *pbp* genes, the relationship between *murM* modifications and penicillin resistance is much less precisely characterized. Therefore, an alignment of the *murM* sequences was analyzed with fastGEAR (*Mostowy et al., 2017b*) to identify any patterns of sequence import from related

species that may be associated with penicillin resistance (*Figure 4—figure supplement 4*). This revealed evidence of recombination with *S. pseudopneumoniae* and *S. mitis* at *murM* in both lineages. However, only one modification, affecting the region between 946 bp and 1143 bp within *murM*, was associated with high-level penicillin resistance. This alteration was observed in both the PMEN9 USA clade and the PMEN3 19A clade, which exhibited the highest penicillin MICs in their respective lineages (*Figure 4—figure supplement 2*).

In PMEN9, there was a high density of recombination events affecting the *dhfR* gene (encoding dihydrofolate reductase; also known as *dyr*), the sequence of which determines resistance to trimethoprim, one of the two components (along with sulfamethoxazole) of co-trimoxazole (*Maskell et al., 2001*). PMEN9 was largely trimethoprim and sulfamethoxazole sensitive, with 60% and 54% of isolates predicted to be susceptible, respectively (*Figure 2*). As with penicillin nonsusceptibility though, resistance to cotrimoxazole components emerged in parallel across multiple clades. Within the South African clade, for instance, 99% were resistant to sulfamethoxazole and 77% were resistant to both trimethoprim and sulfamethoxazole. Even higher levels were observed within the USA and Chinese clades, in which 94% and 100% of isolates were resistant to both trimethoprim and sulfamethoxazole, respectively.

Trimethoprim and sulfamethoxazole resistance were much more widespread among the PMEN3 lineage. In total, 80% of isolates within PMEN3 were trimethoprim resistant and 81% were sulfamethoxazole resistant (*Figure 1*). This spread was mainly driven by the expansion of the ST156 clade, which inherited alleles conferring both these resistance phenotypes. By contrast, within the ST143 clade, only 12% of isolates were trimethoprim resistant and 36% were sulfamethoxazole resistant. The 15 isolates from South Africa in the PMEN3 lineage were all resistant to both trimethoprim and sulfamethoxazole. The high levels of resistance to both antibiotics in South Africa across PMEN3 and PMEN9 could be driven by widespread cotrimoxazole consumption, as it is commonly used as a prophylactic treatment against secondary infections in HIV-positive individuals (*Daniels et al., 2019*).

This was tested using all genomic data from the GPS collection. Most isolates were resistant to sulfamethoxazole (11,594 of 20,043; 58%), with fewer isolates resistant to trimethoprim (7770 of 20,043; 39%). The combination of resistances, conferring full co-trimoxazole resistance, was identified in 7666 isolates (38%). Consistent with the observations within the PMEN3 and PMEN9 lineages, all of these resistance phenotypes were more common in the 4615 South African isolates: 2991 (65%) were resistant to sulfamethoxazole; 2040 (44%) were resistant to trimethoprim; and 1996 isolates (43%) were fully resistant to co-trimoxazole.

## MGE spread in PMEN3 and PMEN9

Other antibiotic resistance phenotypes are determined by acquired genes, rather than alterations to the sequences of core genes. Two types of resistance-associated MGEs were widespread in PMEN3 and PMEN9: those related to Tn*916*, an ICE encoding *tetM* for tetracycline resistance; and those related to Tn*1207.1*, a transposon encoding a *mef(A)/mel* efflux pump causing macrolide resistance (*Johnson and Grossman, 2015*; *Del Grosso et al., 2002*).

Tn*916*-type elements were present in 70 representatives of PMEN3 (*Figure 1*). An ancestral state reconstruction identified 17 independent insertions. Only two spread to a notable extent: one was a clade of 22 isolates within the ST156 clade, and the other was ST143. However, there were multiple instances of Tn*916*-type elements being lost, by 5 and 13 isolates in each clade, respectively. In PMEN9, Tn*916*-type elements were present in 150 isolates (*Figure 2*). The most common was in the South African clade, where Tn*916*-type elements were found in 71 of the 73 isolates in this clade, with likely deletion in two isolates. Similarly, 40 of the 45 isolates within the Chinese clade had also acquired Tn*916*-type elements, with 5 isolates without an element appearing to have lost these independently.

Tn*1207.1*-type elements were more common in both strains. They were found in 108 isolates of PMEN3 (*Figure 1*), resulting from 27 independent insertions. The two insertions associated with the largest clonal expansions were one within the 19A subclade (26 isolates) and a second in another subclade of ST156 (25 isolates). The other 25 insertions were less successful, appearing sporadically around the phylogeny. In PMEN9, Tn*1207.1*-type elements were present in 341 isolates (*Figure 2*). The elements were present in 92 isolates of a subclade of the USA clade and ubiquitous in the 238 isolates of the German clade, the most successful insertion observed in the collection.

Hence, both Tn*916*-type and Tn*1207.1*-type elements were acquired on multiple occasions by both lineages, suggesting frequent importation. However, few of these insertions resulted in internationally disseminated antibiotic-resistant pneumococcal genotypes.

## Selection for local expansion of macrolide-resistant *S. pneumoniae*

The expansion of the German clade carrying Tn*1207.1*-type elements represented an unusual case of an MGE insertion being associated with a successful genotype. This suggested strong selection for a macrolide-resistant genotype in Germany in recent years. However, German antibiotic consumption is generally low relative to the rest of Europe (*Hansen et al., 2013*). Additionally, the German PMEN9 clade is β-lactam susceptible. However, based on macrolide and penicillin consumption data for the period from 1992 to 2010, Germany had a high ratio of macrolide-to-β-lactam usage relative to other European countries (*Figure 5—figure supplement 1*).

All the German isolates within this clade were serotype 14, which was included in the PCV7 vaccine, introduced into the universal vaccination program for children under 2 years of age in Germany in 2006 (*Linden et al., 2016*). Therefore, the 103 isolates collected prior to the introduction of PCV7 were used to test whether this atypical pattern of antibiotic consumption could explain the success of the clade. There was significant evidence of a molecular clock, based on the correlation between the root to tip distance and the date of isolation for this clade (Pearson's correlation coefficient; $R^2$ = 0.15, n = 103, p value<$1 \times 10^{-4}$; *Figure 5—figure supplement 2*). This estimated the clade's MRCA existed in 1970. Generating a time-calibrated phylogeny using BactDating (*Didelot et al., 2018*) suggested a relatively slow clock rate of $5.30 \times 10^{-7}$ substitutions per site per year (95% credibility interval of $3.98 \times 10^{-7}$ to $6.75 \times 10^{-7}$ substitutions per site per year). The Skygrowth package (*Volz and Didelot, 2018*) was then used to reconstruct the effective population size, $N_e$, and the growth rate of $N_e$ through time of this clade. The antibiotic usage data over this period was used as a covariate to test for evidence of selection by changing consumption (*Figure 5*).

From the reconstruction without using the macrolide and β-lactam consumption data, it is evident this lineage expanded rapidly during the late 1990s and early 2000s, with its peak in growth rate around 1997 preceding a peak in $N_e$ around 2003. Both $N_e$ and growth rate subsequently declined. The maximum macrolide-to-β-lactam consumption ratio was in the mid-to-late 1990s. Hence, the peak $N_e$ growth rate, rather than maximum $N_e$ itself, coincided with the timing of the highest consumption ratio. A similar observation was made for methicillin consumption and the spread of methicillin-resistant *Staphylococcus aureus* (*Volz and Didelot, 2018*).

Correspondingly, incorporating the macrolide-to-β-lactam consumption ratio into the reconstruction identified a significant relationship between the lineage's growth rate and this measure of antibiotic consumption. This was reflected by the macrolide-to-β-lactam consumption ratio having a significant positive mean posterior effect of +0.26 (95% credible interval +0.07 to +0.51) on the growth rate of the clade (*Volz and Didelot, 2018*). Additionally, the credible intervals for the growth rate estimation narrowed when consumption data were incorporated into the phylodynamic analysis. Both results support the hypothesis that growth rate was correlated with the contemporary patterns of antibiotic consumption in Germany, consistent with selection pressures from national-level prescribing practices driving the expansion of this clade in the late 1990s. As expected for a β-lactam-sensitive genotype, an analogous analysis found β-lactam consumption by itself had no significant effect on the growth rate of the clade (mean = +0.17, 95% credible interval −0.10 to 0.53; *Figure 5—figure supplement 3*). Macrolide consumption alone did have a significant effect (mean = +0.21, 95% credible interval +0.03 to +0.50; *Figure 5—figure supplement 4*), albeit to a lesser extent than the macrolide-to-β-lactam ratio. Hence, the increasing consumption of macrolides in Germany, compared to β-lactams, can explain this PMEN9 clade's expansion in the 1990s.

The absence of penicillin resistance, or vaccine evasion through serotype switching (*Croucher et al., 2015a*), is a consequence of the Tn*1207.1* element itself. This MGE inserted into, and split, the gene *comEC* (*Figure 5—figure supplement 5*), which encodes a membrane channel protein integral to extracellular DNA uptake during competence (*Bergé et al., 2002*). Therefore, these cells were unable to import DNA for transformation, necessary for serotype switching and the acquisition of penicillin resistance alleles of the *pbp* genes (*von Wintersdorff et al., 2016*). The impact of this insertion is evident from the absence of ongoing transformation within the German clade (*Figure 2*).

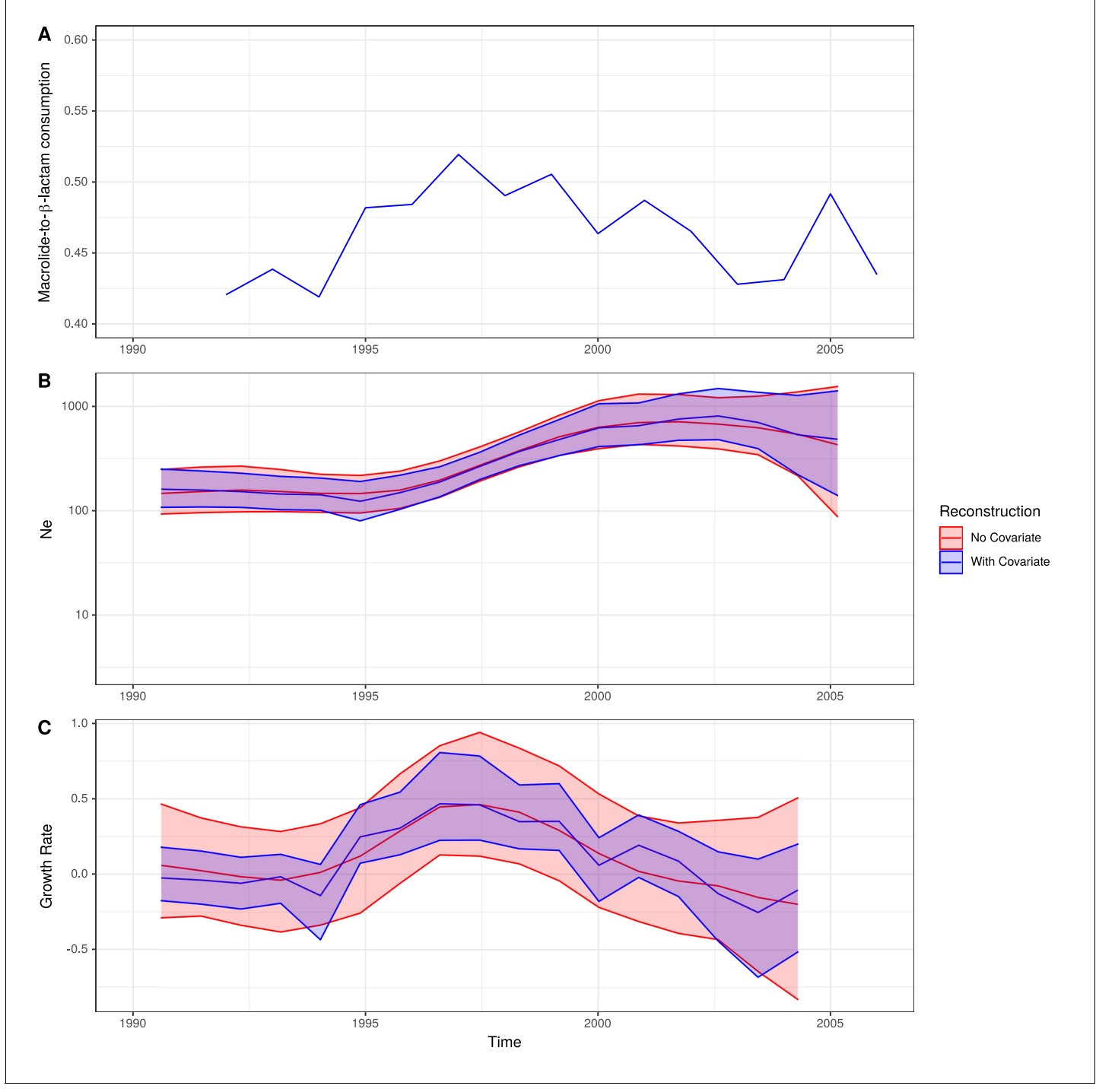

**Figure 5.** Expansion of a macrolide-resistant clade in Germany prior to vaccine introduction. (**A**) The ratio of macrolide-to-β-lactam consumption in Germany. (**B**) The change in $N_e$ through time inferred by Skygrowth, with the red line showing the results of the analysis that did not include covariates, and the blue line showing the results of the analysis that incorporated the macrolide-to-β-lactam ratio into the phylodynamic reconstruction. Shaded regions represent the 95% credible intervals. (**C**) The reconstruction of the growth rate of $N_e$ through time. The red line represents the result of model fitting without covariates, and the blue line when the macrolide-to-β-lactam ratio data were incorporated. Shaded regions represent the 95% credible interval for the reconstruction.

The online version of this article includes the following figure supplement(s) for figure 5:

**Figure supplement 1.** Ratio of macrolide-to-β-lactam consumption in Europe.

**Figure supplement 2.** Root-to-tip analysis of 162 German isolates within PMEN9.

**Figure supplement 3.** Skygrowth analysis incorporating β-lactam consumption data.

*Figure 5 continued on next page*

Analysis of the origin of this MGE, using the flanking regions as for the *pbp* genes above, revealed a probable interspecies origin. The flanking regions immediately adjacent to the insertion have a low percentage identity when aligned to other pneumococci, ranging between 92 and 94% (*Figure 5—figure supplement 5*). The immediate upstream 500 bp region most closely matched to a *S. mitis* reference genome (accession code AFQV00000000). Therefore, other acquisitions of common MGEs, related to either Tn*1207.1* or Tn*916*, were analyzed to determine whether they had also been recently imported from related commensal species.

## Multiple independent acquisitions of resistance genes in *S. pneumoniae*

We first identified the set of insertion sites for Tn*916*-type and Tn*1207.1*-type elements in *S. pneumoniae* using the 20,043 genomes from the GPS project. The genomes were searched for these two elements, and hits were categorized into unique insertion types (a specific combination of MGE length and insertion location) to identify distinct acquisition events. The branch of the GPSC phylogeny on which these insertions occurred was then identified, allowing the determination of whether an element was gained via homologous recombination (see Materials and methods).

At least one of the elements was found in 6101 isolates (30% of the GPS collection) across 262 GPSCs (see Materials and methods). Of these, 1333 isolates contained both Tn*1207.1*-type and Tn*916*-type elements (7%). The Tn*1207.1*-type element was found across 86 GPSCs. The mean prevalence of Tn*1207.1*-type elements in GPSCs in which it was present was 27%. Of the 1971 isolates (10% of the GPS collection) containing Tn*1207.1*-type elements, 1940 isolates were within 146 resistance-associated GPSCs (see Materials and methods). These encompassed 17,590 isolates, upon which further analyses of the insertions were undertaken. For the 1940 isolates in these resistance-associated GPSCs, 1800 (93%) had their insertion point successfully reconstructed. The majority of the 140 isolates where the insertion point was not reconstructed had the Tn*1207.1*-type element present within a small contig with no flanking hits to the reference (74 of 140). There were 50 unique reconstructed insertion types of the Tn*1207.1*-type element, distributed across 27 different insertion loci (*Figure 6*). Some insertion loci were targeted by multiple insertion types. The loci surrounding the *rlmCD* gene, encoding a 23S rRNA methyltransferase, was the most common target, with nine different Tn*1207.1* insertion types targeting this region. The most common insertion type was as a Mega-type cassette within a Tn*916*-like element, which occurred in 1033 (57%) of the isolates. Hence, the diversity of Tn*1207.1* insertion types was relatively low, with a Simpson's diversity index of 0.64.

Tn*1207.1*-type elements sometimes disrupted the host cell's machinery upon their integration. For instance, the second most common insertion type for Tn*1207.1* was the 5.5 kb Mega version of the element inserting into, and splitting, *tag* (*Del Grosso et al., 2006*). The *tag* gene encodes a methyladenine glycosylase, involved in DNA base excision repair. This was present in 260 isolates (14% of identified hits) across 30 different GPSCs. This was also common in the PMEN collections, with Tn*1207.1* within the USA clade of PMEN9 being in the form of Mega splitting *tag* (*Figure 6—figure supplement 1*). The insertion of the 7.2 kb Tn*1207.1* element into *comEC*, as in the German PMEN9 clade, was the third most common, accounting for 5% of insertions (92 isolates) in the GPS collection and appearing in four different GPSCs.

Contrary to the results for PMEN3 and PMEN9, Tn*916*-type elements were more widespread than Tn*1207.1*-types among the collection, being present in 5463 isolates across 248 GPSCs. The mean prevalence for Tn*916* was 63% among GPSCs in which it was present. Of these isolates with Tn*916*-type elements, 5230 were within the 146 resistance-associated GPSCs, upon which further analysis was conducted. The insertion sites of 1895 (36%) of these 5230 isolates were not classifiable. This was primarily due to elements being present in contigs with no, or very short, matches to the reference genome (1496 isolates). For the classifiable 3335 isolates (64% of insertions), there were 407 unique reconstructed insertion types, distributed across 102 different insertion sites (*Figure 7*). The insertion sites harboring the joint greatest number of Tn*916*-type integrations were adjacent to

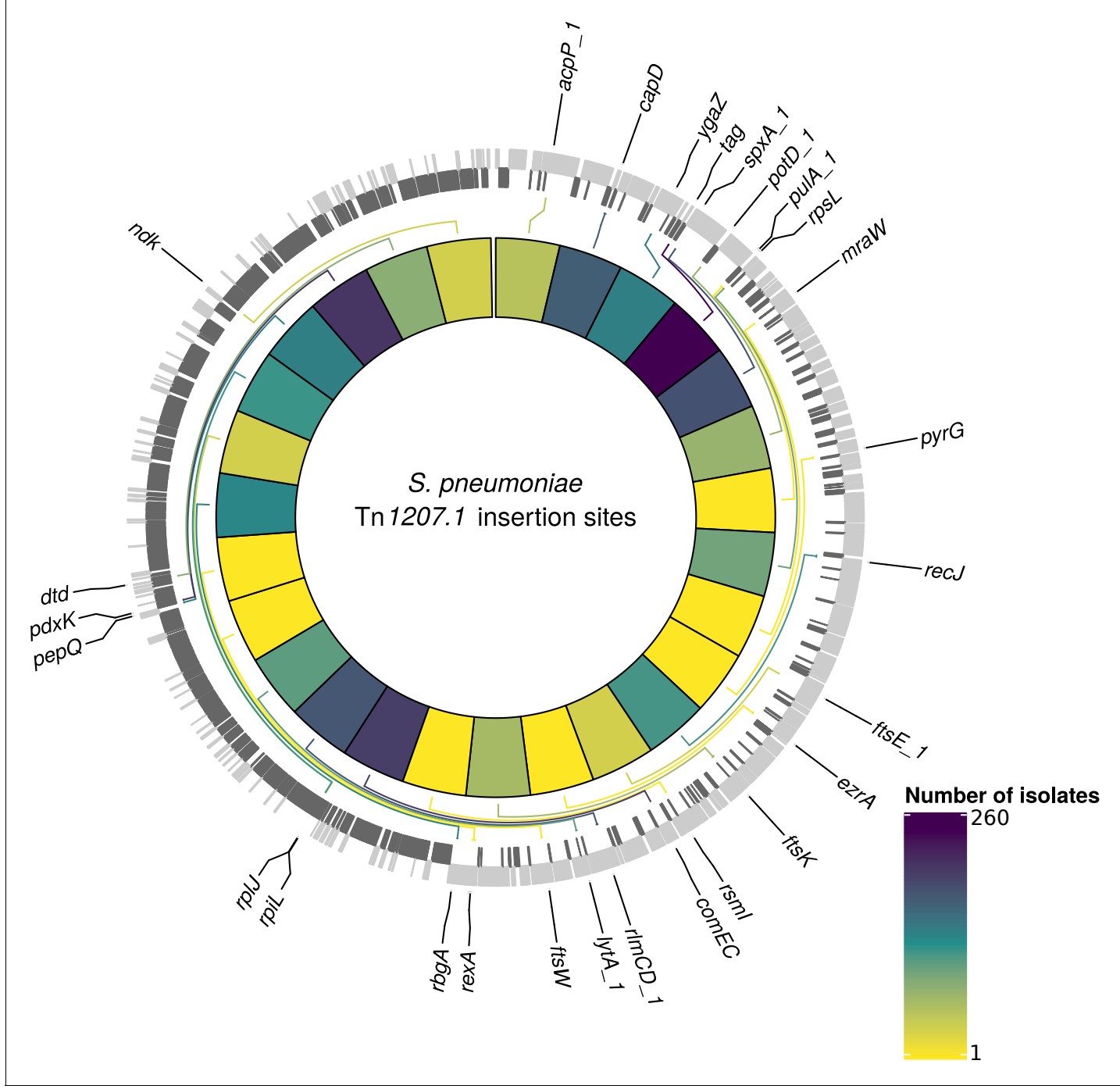

**Figure 6.** Detected insertion points of Tn*1207.1*-type elements within *S. pneumoniae*. Genome of the reference *S. pneumoniae* RMV4 isolate (ENA accession code: ERS1681526) annotated with genes that Tn*1207.1*-type elements have inserted either into, or adjacent to, among the collection. Only genes present within this mobile genetic element (MGE)-free reference are annotated. Gray bars represent coding sequences (CDS): lighter gray bars represent CDS annotated on the forward strand of the genome, darker gray bars represent those on the reverse strand of the genome. The inner heat map represents the number of isolates that have hits inserted into, or adjacent to, the annotated genes. The color scale is logarithmically transformed. The online version of this article includes the following figure supplement(s) for figure 6:

**Figure supplement 1.** Insertion of a Tn*1207.1*-type element as Mega within *tag*.

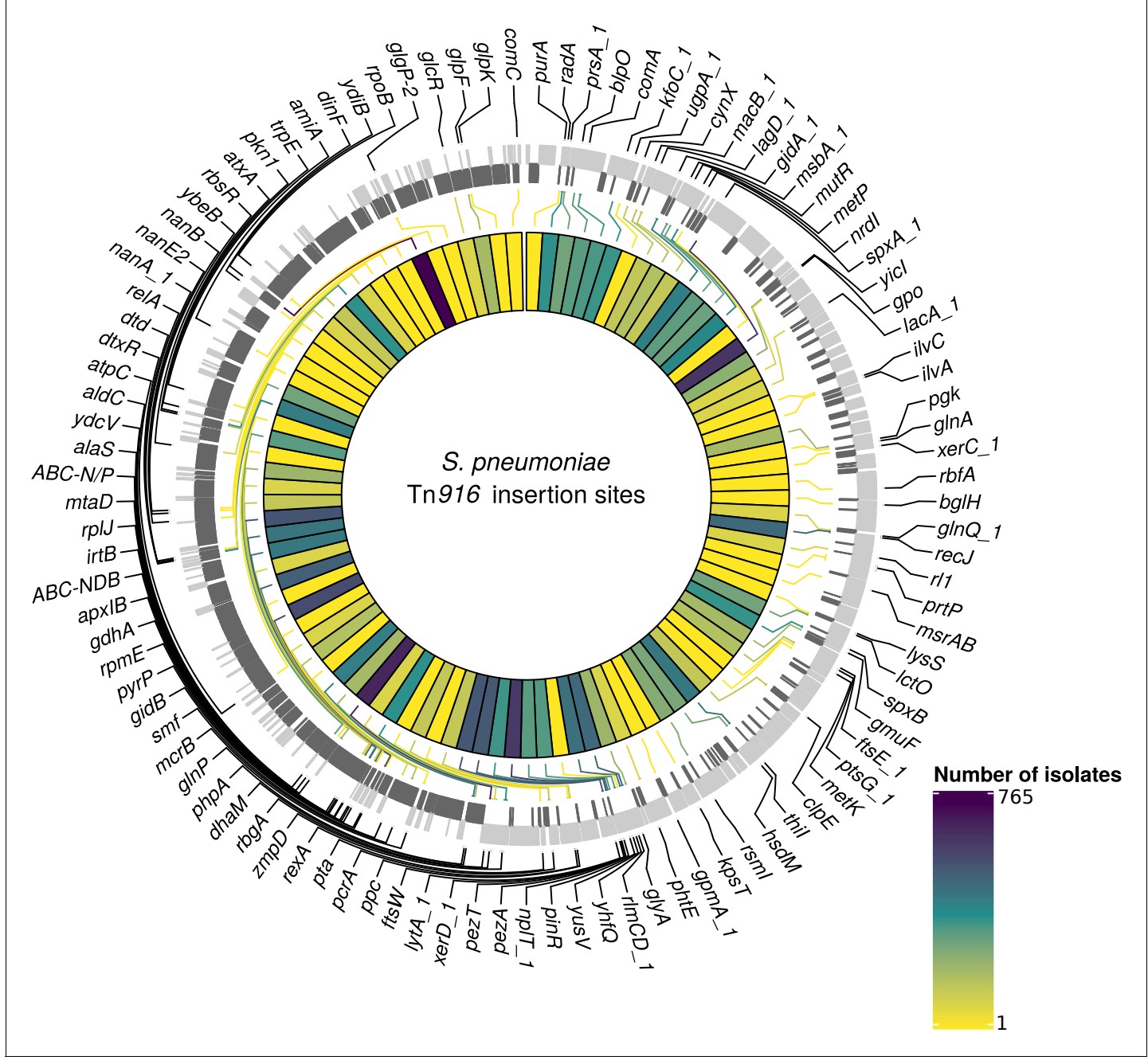

**Figure 7.** Detected insertion points of Tn*916*-type elements within *S. pneumoniae*. Annotated genome of the reference *S. pneumoniae* RMV4 isolate (ENA accession number: ERS1681526) with genes where Tn*916*-type elements have inserted either into, or adjacent to, among the collection. Only genes present within this mobile genetic element (MGE)-free reference are annotated. Gray bars represent coding sequences (CDS): lighter gray bars indicate CDS encoded by the forward strand of the genome, darker gray bars indicate CDS encoded by the reverse strand of the genome. The inner heat map represents the number of isolates that have hits inserted into, or adjacent to, each of the annotated genes. The color scale is logarithmically transformed.

The online version of this article includes the following figure supplement(s) for figure 7:

**Figure supplement 1.** Insertion of a Tn*916*-type element downstream of *recJ*.

**Figure supplement 2.** Insertion of a Tn*916*-type element downstream of *gmuF*.

**Figure supplement 3.** Insertion of a Tn*916*-type element upstream of *gidB*.

**Figure supplement 4.** Insertion of a Tn*916*-type element upstream of *rplL*.

**Figure supplement 5.** Insertion of a Tn*916*-type element upstream of *rplL*.

**Figure supplement 6.** Insertion of a Tn*916*-type element upstream of *rplL*.

the *rbgA* gene, which encodes a ribosomal biogenesis GTPase and is a common insertion site for Tn*5253* (*Santoro et al., 2018*), and the *spxA* gene, which encodes a transcriptional regulator for the *comCDE* competence operon (*Turlan et al., 2009*), with 42 different insertion types proximate to both. The Simpson's diversity index for Tn*916*-type element insertion types was 0.97, indicating these elements are much more variable in how they integrate into the *S. pneumoniae* genome compared to Tn*1207.1*-type elements.

The most common unique reconstructed insertion types for Tn*916*-type elements were insertions of the Tn*2010* and Tn*2009* elements between *ydiB* and *ftsE*, which both encode ATPases involved in cell wall synthesis. These are combinations of Tn*916* and the Mega cassette, related to Tn*1207.1*. The majority of these insertion types occurred within the globally distributed GPSC1 lineage (*Lo et al., 2019*) (containing the PMEN14 lineage): 364 of the 372 Tn*2009* insertion-type examples, and 354 of the 360 Tn*2010* insertion-type examples. Tn*916*-type elements were often present as even larger composite elements, such as the 64.5 kb Tn*5253* element, formed by Tn*916*-type elements inserting into Tn*5252*-type ICE. The next most common insertion type after Tn*2009* and Tn*2010* was a Tn*916*-type element as part of a 66 kb insertion between the immunoglobulin A protease *zmpA* (*Poulsen et al., 1996*) and *rbgA*, present in 177 isolates. This insertion contained the majority of the Tn*5253* backbone, although the Ω*cat* cassette (encoding the chloramphenicol acetyltransferase resistance gene) was missing, and the Tn*916*-type element was Tn*2009* rather than Tn*916* itself. The next most common composite ICE insertion was an 84 kb element containing Tn*2009* and a Ω*cat* element, present in 59 isolates in GPSC16. Overall, Tn*916*-type elements were present in elements over 50 kb in length in 943 isolates (28% of classifiable hits). The diversity in both Tn*5253*-type insertion sites and cassette content made accurately reconstructing Tn*916*-type element insertion types difficult.

## Diverse insertion sites of MGEs

Given this distribution of insertion types and recombination-corrected phylogenies for each strain, ancestral state reconstruction was used to identify the insertions of Tn*916*-type and Tn*1207.1*-type elements across the GPS collection. For Tn*1207.1*-type elements, the 50 unique reconstructed insertion types were found to have inserted 222 times across 59 GPSCs. The most frequent insertion type was the short 4.5 kb element splitting the *tag* gene. This insertion occurred 72 times, representing 32% of all acquisitions of the cassette across *S. pneumoniae.*

For Tn*916*-type elements, there were a much larger number of insertion events: 1023 across the 128 GPSCs in which the insertion types could be reconstructed. Overall, 163 of the 407 Tn*916*-type element insertion types appeared to insert multiple times across the collection. The most frequent insertion (29 times across eight different GPSCs) was a 42 kb Tn*5253*-like element, containing only *tetM* as a resistance gene, inserted upstream of *zmpA*.

The proportion of these insertions occurring within putative recombinations differed between the two elements. For Tn*1207.1*-type elements, 55% of insertions were within recombination blocks (123 of 222) compared with only 8% of the insertions for Tn*916*-type elements (81 of 1023). This difference could have multiple explanations. Tn*916* encodes for its own conjugative machinery and is often present within larger conjugative elements, and therefore may frequently move independently of transformation. Alternatively, Tn*916* may be imported through transformation, but then transpose between loci once in a cell, thus moving away from its site of insertion. Such variation would occur post-insertion, in a pneumococcal strain's recent evolutionary history. The median Simpson's diversity index for within-GPSC Tn*916*-type element insertion site diversity was 0.54, whereas for Tn*1207.1*-type elements it was 0.25. This suggests that Tn*916*-type elements, once inserted, might excise and transpose within the chromosome at a higher rate than Tn*1207.1*-type elements.

Recombinations mediated by transformation are generally much shorter than the lengths of these elements (*Figure 8*), and such exchanges generally favor deletion of elements rather than insertion (*Apagyi et al., 2018*). Comparisons of the length distribution and SNP density for recombination events that imported a Tn*916*-type or Tn*1207.1*-type element, against other recombination events, suggested they were atypical (*Figure 8*). MGE recombinations were significantly longer, with a median length (excluding the length of the element itself) of 10.9 kb, compared to a median length of 7.4 kb for non-MGE recombinations (Mann–Whitney U test; U = 7,775,792, n1 = 183, n2 = 66,419, two-sided p=6.18 $\times$ 10$^{-11}$). Additionally, the median SNP density (excluding the region including the element itself) was significantly higher for MGE recombinations, at 4.41 SNPs per kb,

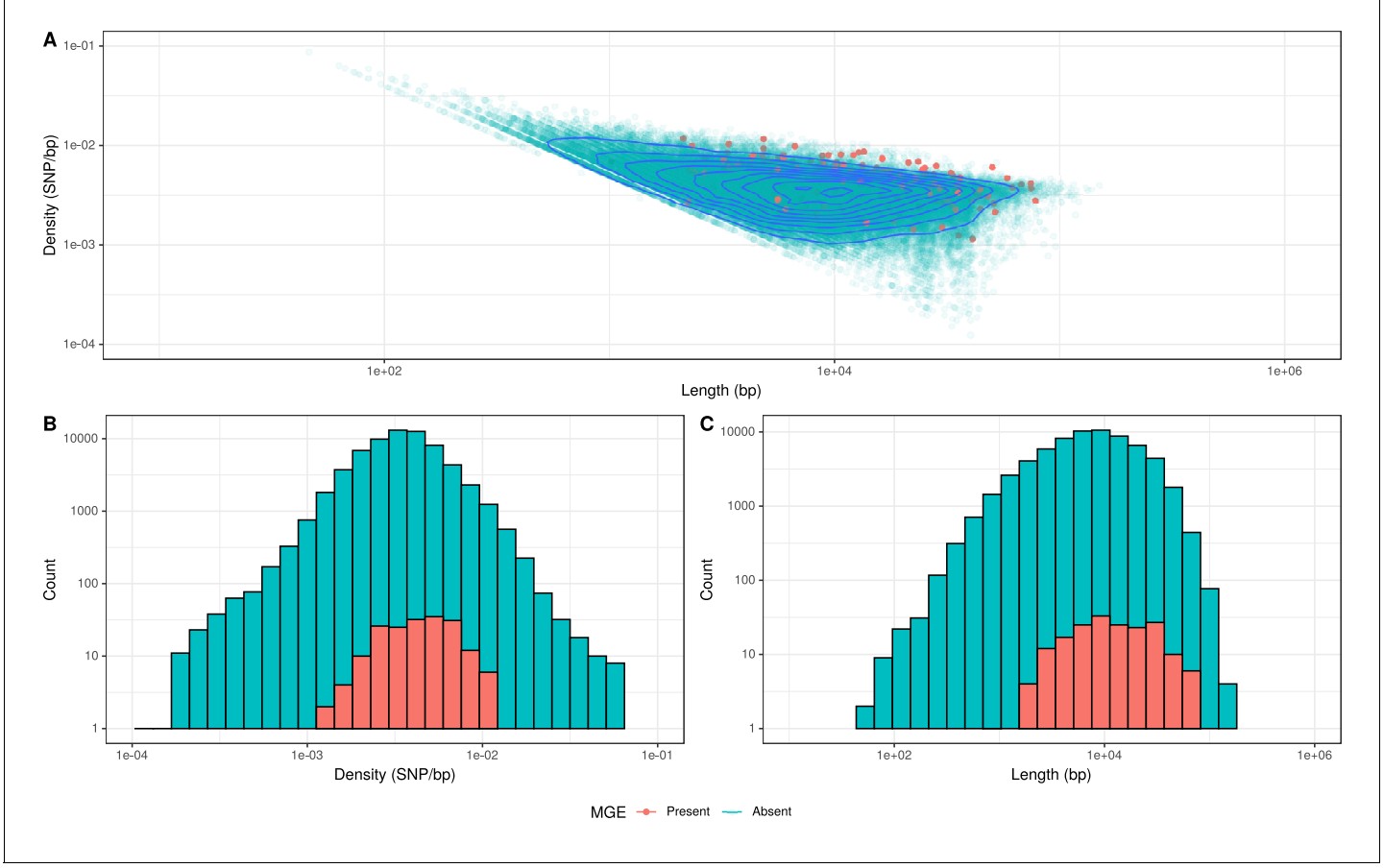

**Figure 8.** Comparison of length and SNP density of recombination events. (**A**) Scatter plot of the SNP density and size of homologous recombinations, with events classified based on whether they imported Tn*916*-type or Tn*1207.10*-type mobile genetic elements (MGEs). Blue contour lines summarize the density of all points. (**B**) Overlaid histogram comparing the SNP density of recombination events that imported MGEs relative to those that did not. (**C**) Overlaid histogram comparing the length of recombinations that imported MGEs relative to those that did not.

compared to non-MGE recombinations with a median of 3.49 SNPs per kb (Mann–Whitney U test; U = 7,643,988, n1 = 183, n2 = 66419, two-sided p=1.62 $\times$ 10$^{-9}$). Given the pneumococcus tends to be conserved at core genome loci (*Lees et al., 2019*), the higher SNP density of these transformation events inserting MGEs was consistent with them originating from donors of other species, as with the integration splitting *comEC* in the German PMEN9 clade.

## Interspecies origin of MGEs

The origin of Tn*1207.1*-type and Tn*916*-type elements imported by homologous recombination was analyzed using the same $\gamma$ score as for the *pbp* loci. For Tn*1207.1*-type elements, the median $\gamma$ score was 0.88 for insertions across the flanking lengths and insertion types. For control isolates, where the element was not inserted and the orthologous flanking regions were extracted, the median $\gamma$ score was 1.0. The overall distribution of $\gamma$ scores was significantly lower between the control and MGE isolates (Mann–Whitney U = 2,860,659, n1 = n2 = 3690, two-sided, p<2.2 $\times$ 10$^{-16}$). This lower score for MGE flanks, relative to orthologous regions in isolates without the MGE, likely represents MGEs being acquired from other species.

For Tn*916*-type element insertions within recombination blocks, the median $\gamma$ scores for both control and MGE isolates were 1.0. However, a Mann–Whitney U test revealed significant difference between the control and MGE isolates $\gamma$ scores, with Tn*916*-type element insertions scoring lower (U = 1,642,284, n1 = 2260, n2 = 2250, two-sided, p<2.2 $\times$ 10$^{-16}$). Hence, there is evidence that some of the Tn*916*-type element insertions occurred through interspecies recombinations.

The trends in the most closely matching species to the flanking regions, over increasing distance from the MGE, followed the expectation for interspecies transfers (*Figure 9*). The control flanking regions matched most closely to pneumococci at all tested lengths. For regions flanking MGE integrations through homologous recombination, non-pneumococcal species matches were much more frequent closer to the insertion. As the flank length increased from 500 bp to 7500 bp, and linkage to the integrated resistance gene decreased, the recombinant isolates were more likely to match pneumococcal DNA.

For Tn*1207.1*-type elements, *S. mitis* was the most likely donor (*Figure 9*). In the regions upstream of the Tn*1207.1* insertion, *S. mitis* was the top match for 92% of 500 bp long flanks. Even at longer flank lengths, *S. mitis* was still the leading match for upstream regions, although for

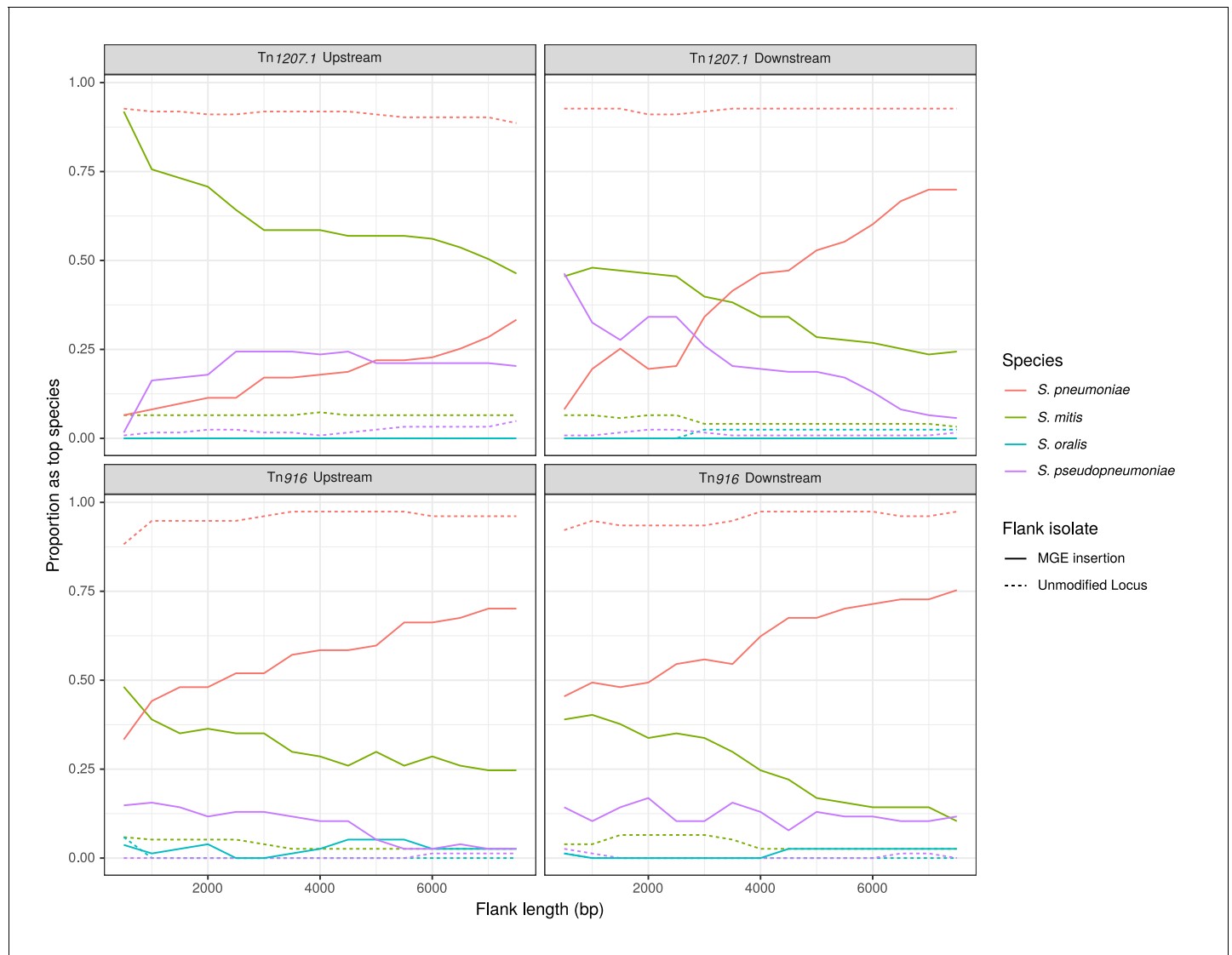

**Figure 9.** Identification of likely sources of Tn*916*-type and Tn*1207.1*-type elements. Flanking regions upstream and downstream from mobile genetic element (MGE) insertion sites were compared to a reference streptococcal database. Lines represent the proportion of matches, across all element acquisitions reconstructed as having occurred through homologous recombination, that correspond to the four species present in the reference streptococcal database. The dashed unmodified locus lines represent data from the orthologous regions of isolates without the MGE insertion. These proportions were calculated at 500 bp increments over 7.5 kb-long flanking regions either side of the insertion site.

The online version of this article includes the following source data and figure supplement(s) for figure 9:

**Source data 1.** Closest species match to the 500 bp region upstream of *tag*-disrupting Tn*1207.1* insertions.
**Figure supplement 1.** Flanking region origin for Tn*1207.1*-type element *tag* insertions.

downstream regions the pneumococcus tended to become the predominant match to flanks by 4 kb outside of the insertion. The most common Tn*1207.1* insertion, that splitting the *tag* gene, can be used to illustrate the local import of sequence from another species (*Figure 9—figure supplement 1*). For the downstream flanks (*Figure 9—figure supplement 1B*), the γ score trend appeared roughly linear with increasing flank length, with the evidence for imported *S. mitis* sequence disappearing 4 kb from the insertion site. However, for the upstream flanking regions (*Figure 9—figure supplement 1A*), the median γ score remained low with increasing flank length, with a median of 0.83 at 7.5 kb upstream of the insertion. This upstream region, replaced by *S. mitis* sequence in many isolates, extended into the *uvrA* gene, another component of the nucleotide excision repair machinery within the pneumococcus. The consistency of top matches for this Tn*1207.1*-type element insertion into *tag* was high across the 66 independent acquisitions within recombination events (*Figure 9—source data 1*). For the 500 bp upstream region, 85% of the insertions (56 of the 66 within recombination blocks) had the *S. mitis* 21/39 (accession code AYRR00000000) reference as their top hit. In total, 97% of these upstream regions (64 of 66) had their top hit as an *S. mitis* sequence.

The less pronounced signal for the interspecies origins of Tn*916*-type element insertions (*Figure 9*) may be a consequence of the difficulty of identifying insertion sites for this larger MGE, resulting in some interspecies transfers being missed. Within the PMEN lineages, 13 of the 61 insertions (21%) of Tn*916*-type elements were detected to have inserted within putative recombination events. All 13 of these had either their immediate upstream or downstream (or both) regions matching most closely to non-pneumococcal species. To verify these insertions were from interspecies recombination events, a selection was also investigated manually. This applied to independent insertions near *recJ* (*Figure 7—figure supplement 1*); *gmuF*, which encodes mannose-6-phosphate isomerase (also known as *manA*; *Figure 7—figure supplement 2*), and *gidB* (*Figure 7—figure supplement 3*). Of these, the upstream regions of the *recJ* and *gmuF* insertions were identified by the algorithm to match most closely to *S. mitis*, while the upstream region of the *gidB* insertion matched most closely to *S. pseudopneumoniae*. The relatively low percent identity scores in the flanking regions inspected manually supports the hypothesis that these elements were likely imported from another species.

Insertions detected outside of putative recombination events were also investigated in detail. Tn*916*-type elements, with flanking fragments of Tn*5252,* inserted near *rplL* on three independent occasions within PMEN3 (*Figure 7—figure supplement 4*; *Figure 7—figure supplement 5*; *Figure 7—figure supplement 6*). This gene encodes a 50S ribosomal protein, and is often the site of Tn*5252*-type ICE integrations (*Croucher et al., 2009*). Given the sequence divergence in the flanking regions from the reference, these insertions were also likely to be interspecies in origin. These were likely missed due to inaccurate reconstruction of the insertion node of these elements, occurring in parallel across the phylogeny. The misclassification of these insertions by the algorithmic approach suggests that these results may underestimate the overall contribution of interspecies homologous recombination in the spread of Tn*916*-type elements.

## Discussion

These analyses describe the evolutionary histories of the *S. pneumoniae* PMEN3 and PMEN9 lineages. Comparisons between the pair illustrate the variable epidemiology of common antibiotic-resistant pneumococci. Most PMEN3 isolates belonged to the penicillin- and cotrimoxazole-resistant ST156 clade, which emerged in the early 1980s, and rapidly spread worldwide. This resembles the rapid global spread of PMEN1 and PMEN14 (*Croucher et al., 2011*; *Croucher et al., 2014c*). However, the antibiotic-resistant bacteria within PMEN9 represent similar resistance profiles emerging independently multiple times, but in clades that remained geographically associated with particular locations. This is despite the GPSC18 strain, from which PMEN9 emerged, originating earlier than PMEN3. Yet, all antibiotic-resistant clades of PMEN3 and PMEN9 shared a history of acquiring resistant loci that had originated from related species via transformation. This points to the important role of commensal species in disseminating resistance genes among pathogenic populations.

As observed in other antibiotic-resistant lineages, both PMEN3 and PMEN9 acquired penicillin resistance via modification of *pbp* genes through the importation of sequence from other species (*von Wintersdorff et al., 2016*; *Dowson et al., 1993*; *Dowson et al., 1990*; *Laible et al., 1991*). This was most frequently observed at the *pbp2b* and *pbp2x* genes, which are usually the first alterations required for low-level penicillin resistance to emerge (*Dewé et al., 2019*). However, substantial

alterations in *pbp1a* are associated with higher levels of resistance, above the 0.12 µg/ml used as the threshold for defining resistance with the RF model (*du Plessis et al., 2002*). Hence, the lack of strong evidence for *pbp1a* being modified by sequence from other species may be an artifact of how transitions between discrete resistance levels were identified in this study. This may also apply to the alterations to the 3′ end of *murM* (*Smith and Klugman, 2001*), modified by integration of sequence from other streptococcal species in PMEN3 and PMEN9 clades exhibiting high penicillin resistance. These clades also had modified *pbp* genes, suggesting epistatic interactions are likely to be important in fully understanding the role of the *murM* imported segments (*Skwark et al., 2017*).

Typically, transformation events generating mosaic *pbp* and *murM* gene structures are short, and confined to few, specific loci (*Hakenbeck et al., 1998*; *Hakenbeck et al., 2001*). Our results extend the importance of interspecies transformation to show its role in importing long stretches of DNA. These long transformation events can cause structural variation, particularly through the integration of antibiotic resistance cassettes, at many sites around the chromosome. Such transformations are atypical in two regards. Firstly, in the high density of SNPs they introduce into the recipient, as the efficiency of exchanges decreases exponentially with sequence divergence (*Majewski et al., 2000*). Secondly, insertion of large loci is rare because transformation events' efficiency decreases exponentially with the length of the imported donor locus (*Apagyi et al., 2018*). Correspondingly, the recombinations importing resistance genes originating in other species are clearly atypical in their properties among all detected homologous recombinations (*Figure 8*). This mirrors the serotype switching recombinations importing the *cps* loci required to escape vaccine-induced immunity, which were much larger than most detected around the genome (*Figure 1—figure supplement 2*). These large *cps* loci recombinations often encompassed the *pbp* loci too. For instance, the emergence of the 19A clade in PMEN3, which had greatly increased penicillin MIC values (*Figure 4—figure supplement 2*), coincided with a large 54 kb recombination spanning the *cps* locus, and the *pbp2x* and *pbp1a* genes, at the base of this clade. Hence, the adaptation of *S. pneumoniae* to medical and public health interventions selects for recombinations with properties that means they are rare, or disruptive, enough to not typically persist in pneumococcal populations.

Therefore, these clinically important, but unrepresentative, recombinations do not provide evidence for the primary evolutionary benefit of transformation (*Redfield, 2001*). Rather, they likely reflect the concept underlying Milkman's hypothesis that exchanges between divergent genotypes will only become common in the recipient where there exists an atypically strong selection pressure (*Milkman et al., 2003*; *Shapiro et al., 2009*). Were they more common, genotypes would routinely converge through recombination (*Fraser et al., 2007*). This suggests the mechanistic or selective barriers to recombinations between streptococcal species in the human oronasopharynx are not absolute, in keeping with the concept of 'fuzzy species' (*Fraser et al., 2007*; *Hanage et al., 2005*; *Sheppard et al., 2008*).

Given the selective pressure to move resistance genes between species, it might have been expected that these exchanges would have occurred through conjugation, rather than atypical transformation events. This is feasible as Tn*916* encodes its own conjugation machinery, while Tn*1207.1* is found within Tn*916*-type ICEs or larger, Tn*5253*-type ICEs. While these large ICEs may impose a burden on the host cell, their site-specific integration machinery is under selection to minimize the disruption of their insertion into the chromosome (*Touchon et al., 2014*). By contrast, transformation's extensive import of sequence from another species flanking the insertion is likely to be deleterious to the recipient. This is especially likely when a host gene is disrupted, as in the example of frequent insertions in the *tag* DNA repair gene, and the integration of Tn*1207.1* into *comEC* in the German PMEN9 clade. This latter example may represent an insertion site that is beneficial for the cassette, rather than the host cell, as the abrogation of competence in this clade would prevent its deletion through subsequent recombination with donors lacking the insertion (*Croucher et al., 2016*). The absence of detected sequence exchanges within this clade highlights the effectiveness of the *comEC* knock out, demonstrating that the competence system mediates most homologous recombination in these PMEN lineages. Furthermore, while selection pressures resulting from local patterns of antibiotic consumption initially enabled the expansion of this clade across Germany during the 1990s and early 2000s, the loss of transformability prevented vaccine evasion through serotype switching, as observed for the local expansion of PMEN2 in Iceland (*Croucher et al., 2014a*).

Nevertheless, HGT mechanisms other than transformation still contributed to the dissemination of Tn*1207.1*-type and Tn*916*-type elements. Transformation was estimated to make a larger

contribution to the spread of the smaller Tn*1207.1*-type elements relative to the larger Tn*916*-type elements. The short length of Tn*1207.1* both makes it easier to identify the insertion site of Tn*1207.1*-type elements in contigs from draft assemblies (1239 Tn*916* insertions were present in contigs lacking sufficient matches to a reference, as opposed to 101 for Tn*1207.1*), and makes it more likely that Tn*1207.1*-type elements can be moved by transformation (*Apagyi et al., 2018*). Furthermore, Tn*916*-type elements encode machinery for transposition, including an integrase from the transposase subfamily of tyrosine recombinases (*Roberts and Mullany, 2009*). This integrase exhibits little sequence specificity in its insertion site preference, although it generally favors sites that are AT-rich or bent (*Roberts and Mullany, 2009*; *Wozniak and Waldor, 2010*). Hence even when imported by a transformation event, Tn*916*-type elements may be able to move within the chromosome, and thereby disassociate themselves from any imported flanks more efficiently than Tn*1207.1*-type elements, which lack such machinery. This intragenomic mobility likely also accounts for the Tn*916*-type elements being observed to insert at over 100 locations in the pneumococcal genome, whereas Tn*1207.1*-type elements were only found at 27 sites.

Precisely determining the species of origin for observed interspecies recombination events is challenging, given the diversity of oral streptococci and their ability to exchange sequence. While our reference database is sufficient to split likely non-pneumococcal from pneumococcal DNA, it is not detailed enough to fully delineate the networks through which AMR genes spread. Though *S. mitis* is clearly a crucial source of antibiotic resistance genes for *S. pneumoniae*, the much greater diversity of this commensal means the few available samples are spread thinly across the population structure (*Kilian et al., 2014*). As such, greater sampling of more streptococcal species is needed to assess the most likely donor for these interspecies transformations. More fully describing the genetic diversity of commensals may enable an improved understanding of what proportion of resistance loci can move across the species boundaries into pathogens.

In conclusion, this study has identified the broader importance of interspecies transformation in the emergence of antibiotic-resistant *S. pneumoniae*. The atypical properties of these large and SNP-dense events underscore the strength of selection for adaptive evolution resulting from clinical interventions relative to naturally occurring pressures. This suggests that there is a continual flow of sequence between related species sharing a niche, which is normally inhibited by outbreeding depression (*Harrow et al., 2021*), but may enable rapid adaptation following public health interventions against pathogens. Such transfers are sufficiently frequent for resistant genotypes to emerge and spread locally, but particularly successful genotypes, such as PMEN3, can rapidly spread between continents. This highlights the challenges of blocking the transfer of resistance loci into pathogenic species.

# Materials and methods

## Bacterial isolates and DNA sequencing

Isolates belonging to the *S. pneumoniae* PMEN3 and PMEN9 lineages were collated from across Europe (from the Nationales Referenzzentrum für Streptokokken, Germany; and the collections of Prof. de Lencastre), the Americas (from the collections of the CDC through the Global Strain Bank Project), and the Maela refugee camp in Thailand (*Turner et al., 2012*; *Figure 1—source data 1*). These collections were associated with MLST data (*Enright and Spratt, 1998*). Therefore, isolates of sequence type (ST) 156, and single locus variants thereof, could be selected as representatives of PMEN3 (or Spain[9V]-3); isolates of ST9, and single locus variants thereof, were selected as representatives of PMEN9 (or England[14]-9) (*McGee et al., 2001*). This generated collections of 272 and 325 isolates for PMEN3 and PMEN9, respectively. Isolates that could be cultured were sequenced as paired-end 24plex libraries on Illumina HiSeq 2000 machines, generating 75-nt reads. Sample identity was checked through comparing serotype, inferred by seroba v1.0.0 (*Epping et al., 2018*), and ST with those determined by sample providers. Samples were checked for contamination through assessing their mapping to the reference sequence, as described previously (*Croucher et al., 2011*). After these tests, 215 PMEN3 and 263 PMEN9 isolates passed for use in the described analyses.

These datasets were combined with isolates from the GPS project, which generated a database of 20,043 high-quality pneumococcal draft genome sequences from 33 countries collected between 1991 and 2017 (*Gladstone et al., 2019*). In total, 49.7% of these isolates were collected from

locations before the PCV7 vaccine was introduced. Most isolates were sampled from cases of IPD in children under the age of five. PMEN3 and PMEN9 corresponded to strains GPSC6 (454 isolates associated with information on country and date of isolation) and GPSC18 (312 isolates associated with information on country and date of isolation) in this collection, respectively (*Lees et al., 2019*). Hence, the final dataset sizes were 669 isolates for GPSC6, containing PMEN3, and 575 isolates for GPSC18, containing PMEN9. Among these PMEN collections, 64.7% of isolates were collected before the PCV7 vaccine was introduced. Raw and processed sequence data for the 478 isolates not within the GPS collection are publicly available in the EMBL Nucleotide Sequence Database (ENA; project number PRJEB2255). All accession codes for reads, assemblies, and annotations are listed in *Figure 1—source data 1*.

Across the wider GPS collection, 146 GPSCs were identified as 'resistance associated'. These GPSCs contained more than 10 isolates in total, at least one of which encoded either Tn*1207.1* or Tn*916*. The 17,590 isolates in these resistance-associated GPSCs were used in the phylogenetic analyses described below.

## Generation of annotation and alignments

De novo assemblies were generated using an automated pipeline for Illumina sequences (*Page et al., 2016*). Briefly, reads were assembled using Velvet with parameters selected by VelvetOptimiser. These draft assemblies were then improved by using SSPACE and GapFiller to join contigs (*Zerbino and Birney, 2008*; *Boetzer and Pirovano, 2014*; *Boetzer and Pirovano, 2012*). The final assemblies were annotated using PROKKA (*Seemann, 2014*).

Whole-genome alignments were generated for phylogenetic analysis through mapping of short read data against reference sequences. For the PMEN3 and PMEN9 analyses, the reference genomes were *S. pneumoniae* RMV4 *rpsL** Δ*tvrR* (accession code: ERS1681526) (*Kwun et al., 2018*) and INV200 (accession code: FQ312029.1), respectively. Mapping was performed using SMALT v0.64, the GATK indel alignment toolkit, and SAMtools as described previously (*Croucher et al., 2012*). A faster method was applied to GPSCs containing more than 10 isolates in the GPS study. A reference sequence was chosen as the isolate with the largest $N_{50}$ value (the length of the contig at the midpoint of the assembly, when contigs are ordered by size). Other isolates were mapped to this reference using SKA (*Harris, 2018*) with default settings.

## Antibiotic consumption data

Selection pressures on specific clades were inferred using macrolide and penicillin consumption data from Europe. Two data sources were used for macrolides: a study looking at macrolide resistance among pneumococci isolates in Germany by *Reinert et al., 2002*, which recorded data from 1992 to 2000; and the European Centre for Disease prevention and Control (ECDC), which recorded data from 1997 to the present day. The ECDC data for Germany is from the primary care sector for outpatients, with a population coverage of 90%, while the Reinert et al. paper takes data from both prescriptions in hospitals and from community general practitioners. The macrolide usage data were combined using the 3 years of overlap between the two datasets as a scaling factor. This was the mean transformation that mapped the *Reinert et al., 2002* data to the ECDC data. It was applied to convert the data from 1992 to 1996 into the same units as the ECDC data (defined daily doses per 1000 individuals in the population).

For β-lactam consumption, data were also taken from the ECDC for 1997 to present day. For β-lactam consumption from 1992 to 1997, data were taken from *McManus et al., 1997*. These data reflect hospital and retail sales of oral antibiotics in West Germany for the years 1989 and 1994 in the same DDD units as the ECDC data. A linear trend between 1989, 1994, and 1997, the first year of data from the ECDC, was used to impute the missing values between 1992 and 1997.

## Phylogenetic and phylodynamic analyses

Gubbins v2.3 (*Croucher et al., 2015b*) was used to identify recombinations and generate phylogenies for both the PMEN lineages and the GPSCs. Gubbins was run for five iterations. The starting phylogeny of isolates was constructed with FastTree 2 (*Price et al., 2010*). Subsequent iterations generated phylogenies with RAxML v8.2.8 (*Stamatakis, 2014*), with a generalized time reversible (GTR) model of nucleotide substitution with a discretized gamma distribution of rates across sites.

Time-calibrated phylogenies were generated from the Gubbins outputs using the BactDating R package v1.0.1 (*Didelot et al., 2018*). Isolates without dates of collection were pruned from the phylogeny, and the root-to-tip distances used to test for a molecular clock signal. Where one was detectable, BactDating was run with a relaxed clock model and a Markov chain Monte Carlo (MCMC) length of 50 million iterations. Chain convergence was checked through visual inspection of trace plots.

The Skygrowth R package (*Volz and Didelot, 2018*) was then used to formally test the link between antibiotic consumption and population growth rates. The timed phylogeny generated by BactDating was input into Skygrowth, where it was analyzed in combination with the β-lactam, macrolide, and macrolide-to-β-lactam consumption data. Consumption data were each separately rescaled prior to analysis with default settings and priors. Four sets of analysis were run: one for each of the consumption datasets, and one without consumption data. In each case, the MCMC was run for 100 million iterations, which visual inspection suggested was sufficient for the chains to converge. β-lactam consumption data alone had no significant effect on the reconstruction, while macrolide consumption had a smaller effect than the macrolide-to-β-lactam consumption data.

## Antibiotic resistance analyses

The MIC for penicillin had been determined for most isolates in the PMEN3 and PMEN9 collections (65.5 and 80.9%, respectively). The MICs of the remaining 341 isolates were predicted using an RF approach analogous to that developed in *Li et al., 2017*. In this approach, the TPD of three penicillin binding proteins (PBPs; PBP1A, PBP2B, and PBP2X) were extracted and each amino acid position used as a predictor to train an RF model on the continuous $\log_2$ MIC value. The training data came from 4342 isolates previously characterized by the *CDC, 2018*.

This model predicted the MIC for all 341 isolates with unknown MIC values. The continuous MIC values predicted by the model were then converted into categories based on the pre-2008 CLSI meningitis breakpoints for resistance, with an added intermediate class of isolates for those with 0.06 μg/l < MIC < 0.12 μg/l (*Centers for Disease Control and Prevention (CDC), 2008*). This method was then used on the wider 20,043 GPS collection, with 59 isolates (0.3%) having unidentifiable *pbp* genes.

Resistance to sulfamethoxazole was detected using a hidden Markov model (HMM), constructed using HMMer3 (*Eddy, 2011*), trained to extract the region downstream of S61 in *folP*. If this region contained at least one inserted amino acid, then the isolate was predicted to be resistant. Resistance to trimethoprim used another HMM to identify the amino acid at position 100 in *dhfR* (also known as *dyr* or *folK*). Isolates with an isoleucine at this position were predicted to be sensitive, and isolates with a leucine at this position were assumed to be resistant (*Adrian and Klugman, 1997*). If isolates were classified as resistant to both sulfamethoxazole and trimethoprim, they were also classified as resistant to the combination drug cotrimoxazole (*Metcalf et al., 2016*).

The code for both the penicillin prediction and cotrimoxazole prediction methods is available at https://github.com/jdaeth274/pbp_tpd_extraction (copy archived at swh:1:rev:353ed3fad766fe-c21a011301ebc49c3fe356c305, *D'Aeth, 2021a*).

## Ancestral state reconstruction

The penicillin resistance categories inferred from the metadata and the RF model were reconstructed on the time-calibrated phylogeny using the phytools R package v0.7.7 (*Revell, 2012*). The make.simmap function was run using an equal rates model and an MCMC chain sampling every 100 iterations. The input was a matrix of character states for the tips, in which each isolate's observed or predicted phenotypes were assigned a probability of 1.

After the reconstruction, each node's state was assigned as that with the highest posterior probability. Starting at the root, the number of lineages of each state at each coalescent event in the time-calibrated tree was recorded. Every time a node was reached, an extra lineage was added to the total. If there was no state change between two nodes, the count for the state was increased by 1; if there was a state change not on a terminal branch, the count for the new state was increased by 2; else if there was a state change on a terminal branch, the new state count was increased by 1. The total number and the proportion of branches in each state were recorded through time.

To assess the number of serotype switching events across the PMEN lineages, the JOINT maximum likelihood model for ancestral reconstruction of PastML v1.9.15 (*Ishikawa et al., 2019*) was used.

## MGE identification

Reference MGE sequences were used to search all the genomic datasets for intact and partial representatives. For the Tn*916* element, the 18 kb reference given by the transposon registry (*Tansirichaiya et al., 2019*), extracted from *Bacillus subtilis* (accession code: KM516885), was used. For Tn*1207.1*, a 7 kb reference extracted from the *S. pneumoniae* INV200 genome (accession code: FQ312029.1) was used. BLASTN was used to detect Tn*916* and Tn*1207.1* among the assembled genomes in the collections, with hits filtered using an empirically determined tthreshold alignment length of 7 kb and 2 kb, respectively, for each element. BLASTN results were merged if they represented continuation of an element's sequence split across multiple contigs, to enable detection of elements in isolates that were fragmented in draft assemblies.

## MGE insertion site identification

A pipeline was developed to categorize the insertion points of the elements and infer the node at which the insertion occurred within a cluster's phylogeny. *Figure 10* outlines the algorithm. The initial step was the creation of a library of unique hits, with BLAST matches against the GPSC's reference genome determining the start and end points of an insertion. A hit was defined by three characteristics: (i) the total length of the delineated insertion, (ii) the number of genes within the insertion, and (iii) the genes within the flanking regions of a hit. Each observed combination of values was considered a unique hit. For instance, if two hits were of similar length and gene content, but differed in where they inserted within the host, they were treated as two unique hits. The unique insertions with the longest flanking matches to the reference, indicating the insertion was detected on a large contig, were used as representatives of that insertion within the library. Hits that had either shorter flanking sequences, or an MGE insertion spread across contigs, were not considered as candidates to be a library hit. The next step was to allocate the remaining hits, not present in the

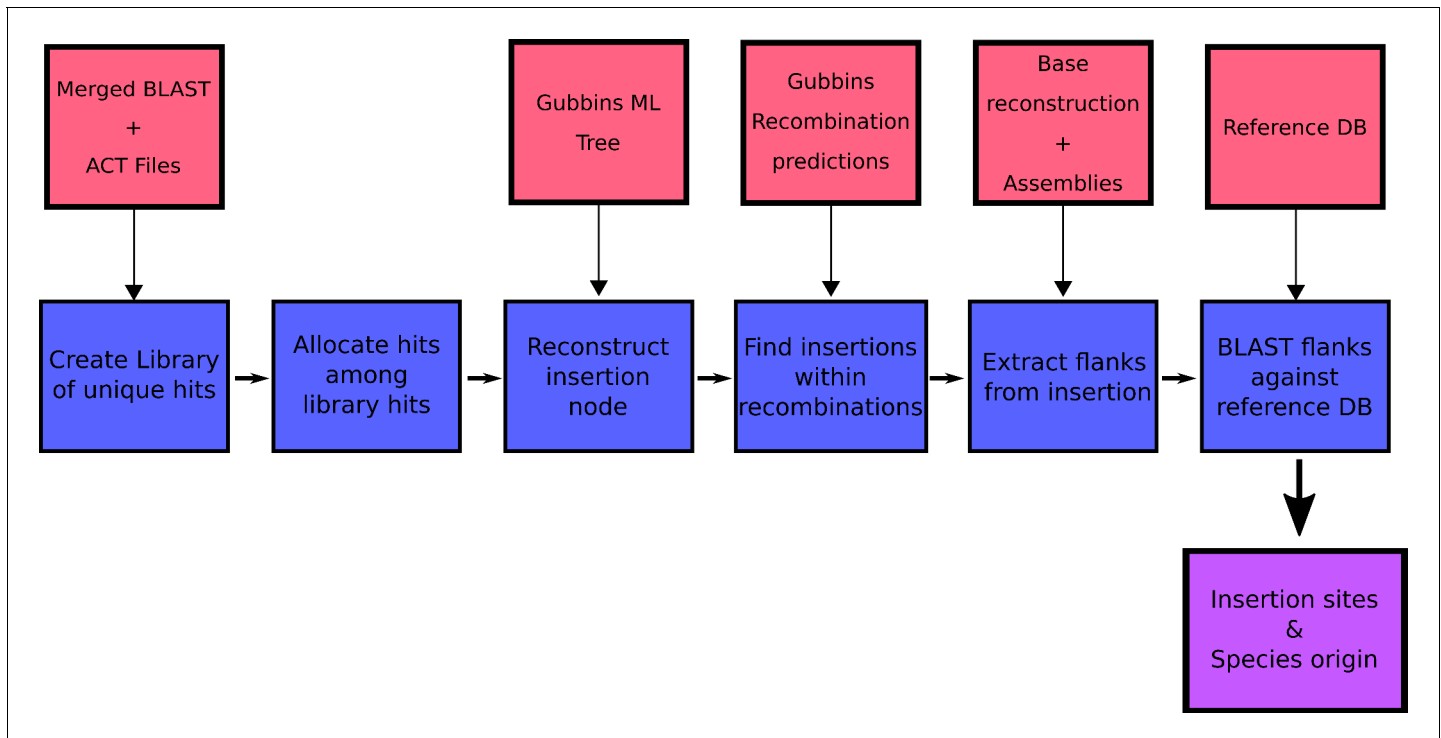

**Figure 10.** Overview of pipeline used to analyze the acquisition of Tn*916*-type and Tn*1207.1*-type elements. Red boxes represent data input into the pipeline, blue boxes the individual analysis steps within the pipeline, and purple the pipeline's output.

library, to one of the unique library insertion types (the combination of gene number, insertion length and location). Isolates with no matches to the reference either side of the hit, usually when the hit was present in a small contig or within a larger unresolved element, were discarded from the analysis. Once hits had been allocated an insertion type, the node at which the insertion occurred was reconstructed on the Gubbins phylogeny for each GPSC. This ancestral state reconstruction was performed using PastML (*Ishikawa et al., 2019*), as these phylogenies were not time-calibrated. The recombination predictions were then searched to detect whether there was a putative recombination event, on the branch on which acquisition was estimated to occur, spanning the insertion site within the reference for a GPSC. If such a recombination were identified, this was considered indicative of element insertion via homologous recombination. The flanking regions of the isolate with the fewest reconstructed SNPs around the insertion site of the element since its insertion, as inferred from the Gubbins base reconstruction, were then extracted to test for the origin of this element.

These flanking regions were compared to a reference collection of 52 streptococcal genomes collated from antimicrobial-susceptible *S. pneumoniae* and other *Streptococcus* species, building on the database collated in *Mostowy et al., 2017b*. BLASTN was used to compare each flanking region to this database. The orthologous regions to these flanks were also extracted from isolates not containing the insertion, to act as a control.

The statistic $\gamma$ was used to determine the species of origin for an insertion. This used the BLAST bit score, which is a normalized form of the raw score of an alignment. The bit score measures sequence similarity irrespective of query sequence length and database size. The $\gamma$ statistic was calculated as the bit score of the top-ranked *S. pneumoniae* hit (*b*) divided by the bit score of the top-ranked hit (*B*):

$$\gamma = \frac{b}{B} \tag{1}$$

Hits where the top match was *S. pneumoniae*, indicating the insertion originated from an intraspecies transformation event, had a $\gamma$ score of 1.0. Any score below 1.0 indicated a potential origin from outside of *S. pneumoniae*.

This pipeline was also applied to detect the origin of *pbp* genes involved in the acquisition of resistance. Here, using phenotype predictions from the RF model described above, ancestral node resistance states were reconstructed. The descendants of nodes where resistance was acquired, or lost, with the fewest base substitutions subsequently accumulating in the three *pbp* genes then had their gene sequences extracted. These sequences were then compared to the reference database using BLASTN. The same $\gamma$ statistic as above was used to detect the likely origin of these *pbp* genes. The code for both the altered *pbp* pipeline and the MGE detection pipeline is available at https://github.com/jdaeth274/ISA (copy archived at swh:1:rev:c3873d851fdfb01efd8bb1f8a18f33acb06b6fc5, *D'Aeth, 2021b*).

For *murM*, where the effects of alterations on resistance levels are less well understood, a different approach was taken. The regions corresponding to the *murM* genes in the annotated references were extracted from the PMEN3 and PMEN9 whole-genome alignments. To enable the detection of possible interspecies recombinations, *murM* sequences from *S. mitis* 21/39 (accession code: AYRR01000000) and *Streptococcus pseudopneumoniae* IS7493 (accession code: CP002925) were added to the dataset. All *murM* sequences were then aligned with Muscle v3.8.31 (*Edgar, 2004*). Sequences were clustered into lineages, and recombinations inferred, using fastGEAR (*Mostowy et al., 2017b*).

## Acknowledgements

NJC and JCD were supported by the UK Medical Research Council and Department for International Development (grant nos. MR/R015600/1 and MR/T016434/1). NJC was supported by a Sir Henry Dale Fellowship, jointly funded by Wellcome and the Royal Society (grant no. 104169/Z/14/A). JCD also acknowledges PhD funding from the Wellcome Trust (grant no. 102169/Z/13/Z). The GPS study was cofunded by the Bill and Melinda Gates Foundation (grant code OPP1034556), the Wellcome Sanger Institute (core Wellcome grants 098051 and 206194), and the US Centers for Disease Control and Prevention. WPH was supported by the National Institutes of Health (grant number R01 AI106786). PT was supported by the Wellcome trust (grant number 106698).

# Additional information

## Group author details

**The GPS Consortium**

Alejandra Corso: Administración Nacional de Laboratorios e Institutos de Salud, Buenos Aires, Argentina; Diego Faccone: Instituto Nacional de Enfermedades Infecciosas (INEI)-Administración Nacional de Laboratorios e Institutos de Salud (ANLIS) "Dr. C. Malbrán", Buenos Aires, Argentina; Paula Gagetti: Administración Nacional de Laboratorios e Institutos de Salud, Buenos Aires, Argentina; Abdullah W Brooks: International Centre for Diarrheal Diseases Research, Dhaka, Bangladesh; Md Hasanuzzaman: Child Health Research Foundation, Dhaka, Bangladesh; Roly Malaker: Child Health Research Foundation, Dhaka, Bangladesh; Samir K Saha: Child Health Research Foundation, Dhaka, Bangladesh; Alexander Davydov: The Republican Research and Practical Center for Epidemiology and Microbiology, Minsk, Belarus; Leonid Titov: The Republican Research and Practical Center for Epidemiology and Microbiology, Minsk, Belarus; Maria Cristina de Cunto Brandileone: Center of Bacteriology, Adolfo Lutz Institute, São, Paulo, Brazil; Samanta Cristine Grassi Almeida: Center of Bacteriology, Adolfo Lutz Institute, São, Paulo, Brazil; Margaret Ip: Chinese University of Hong Kong, Hong Kong, China; Pak Leung Ho: The University of Hong Kong, Hong Kong, China; Pierra Law: The University of Hong Kong, Hong Kong, China; Chunjiang Zhao: Peking University People 's Hospital, China; Hui Wang: Peking University People 's Hospital, China; Jeremy Keenan: University of California, San Francisco, San Francisco, United States; Eric Sampane-Donkor: School of Biomedical and Allied Health Sciences University of Ghana, Accra, Ghana; Balaji Veeraraghavan: Christian Medical College, India; Geetha Nagaraj: Kempegowda Institute of Medical Sciences Hospital & Research Center, Bangalore, India; KL Ravikumar: Kempegowda Institute of Medical Sciences Hospital & Research Center, Bangalore, India; Noga Givon-Lavi: Ben-Gurion University of the Negev, Beer-Sheva, Israel; Nurit Porat: Ben-Gurion University of the Negev, Beer-Sheva, Israel; Rachel Benisty: Ben-Gurion University of the Negev, Beer-Sheva, Israel; Ron Dagan: Ben-Gurion University of the Negev, Beer-Sheva, Israel; Godfrey Bigogo: Kenya Medical Research Institute, Kisumu, Kenya; Jennifer Verani: Centers for Disease Control and Prevention, Atlanta, United States; Anmol Kiran: Malawi-Liverpool-Wellcome-Trust, Blantyre, Malawi; Dean B Everett: University of Edinburgh, Edinburgh, United Kingdom; Jennifer Cornick: Malawi-Liverpool-Wellcome-Trust, Blantyre, Malawi; Maaike Alaerts: Malawi-Liverpool-Wellcome-Trust, Blantyre, Malawi; Shamala Devi Sekaran: MAHSA University, Selangor, Malaysia; Stuart C Clarke: University of Southampton, Southampton, United Kingdom; Houria Belabbès: Ibn Rochd university-hospital center, Casablanca, Morocco; Idrissa Diawara: Mohammed VI University of Health Sciences (UM6SS), Casablanca, Morocco; Khalid Zerouali: Faculté de médecine et de pharmacie de Casablanca, Casablanca, Morocco; Naima Elmdaghri: Faculty of Medicine and Pharmacy & Ibn Rochd University Hospital Center, Casablanca, Morocco; Benild Moiane: Centro de Investigação em Saúde da Manhiça, Maputo, Mozambique; Betuel Sigauque: Centro de Investigação em Saúde da Manhiça, Maputo, Mozambique; Helio Mucavele: Centro de Investigação em Saúde da Manhiça, Maputo, Mozambique; Andrew J Pollard: University of Oxford and the NIHR Oxford Biomedical Research Centre, Oxford, United Kingdom; Rama Kandasamy: University of Oxford and the NIHR Oxford Biomedical Research Centre, Oxford, United Kingdom; Philip E Carter: Kenepuru Science Centre, Porirua, New Zealand; Stephen Obaro: University of Nebraska Medical Center, Omaha, United States; Sadia Shakoor: The Aga Khan University, Karachi, Pakistan; Deborah Lehmann: The University of Western Australia, Perth, Australia; Rebecca Ford: Papua New Guinea Institute of Medical Research, Goroka, Papua New Guinea; Theresa J Ochoa: Instituto de Medicina Tropical, Universidad Peruana Cayetano Heredia, Lima, Peru; Anna Skoczynska: National Medicines Institute, Warsaw, Poland; Ewa Sadowy: National Medicines Institute, Warsaw, Poland; Waleria Hryniewicz: National Medicines Institute, Warsaw, Poland; Sanjay Doiphode: Hamad Medical Corporation, Doha, Qatar; Ekaterina Egorova: Moscow Research Institute for Epidemiology and Microbiology, Moscow, Russian Federation; Elena Voropaeva: Gabrichevsky Epidemiology and Microbiology Research Institute, Moscow, Russian Federation; Yulia Urban: Gabrichevsky Epidemiology and Microbiology Research Institute, Moscow, Russian Federation; Metka Paragi: National

Laboratory of Health, Environment and Food, Ljubljana, Slovenia; Tamara Kastrin: National Laboratory of Health, Environment and Food, Ljubljana, Slovenia; Anne Von Gottberg: Centre for Respiratory Diseases and Meningitis, National Institute for Communicable Diseases, Johannesburg, South Africa; Kedibone M Ndlangisa: Centre for Respiratory Diseases and Meningitis, National Institute for Communicable Diseases, Johannesburg, South Africa; Linda De Gouveia: Centre for Respiratory Diseases and Meningitis, National Institute for Communicable Diseases, Johannesburg, South Africa; Mignon Du Plessis: Centre for Respiratory Diseases and Meningitis, National Institute for Communicable Diseases, Johannesburg, South Africa; Mushal Ali: Centre for Respiratory Diseases and Meningitis, National Institute for Communicable Diseases, Johannesburg, South Africa; Nicole Wolter: Centre for Respiratory Diseases and Meningitis, National Institute for Communicable Diseases, Johannesburg, South Africa; Shabir A Madhi: University of the Witwatersrand, Johannesburg, South Africa; Susan A Nzenze: University of the Witwatersrand, Johannesburg, South Africa; Somporn Srifuengfung: Faculty of Medicine, Siam University, Thailand; Faculty of Medicine Siriraj Hospital, Mahidol University, Thailand; Brenda Kwambana-Adams: Medical Research Council Unit, Banjul, Gambia; Ebenezer Foster-Nyarko: Medical Research Council Unit, Banjul, Gambia; Ebrima Bojang: Medical Research Council Unit, Banjul, Gambia; Martin Antonio: Medical Research Council Unit, Banjul, Gambia; Peggy-Estelle Tientcheu: Medical Research Council Unit, Banjul, Gambia; Jennifer Moïsi: Agence de Médecine Préventive, Paris, France; Michele Nurse-Lucas: Department of Paraclinical Sciences, The University of the West Indies, St., Augustine, Trinidad and Tobago; Patrick E Akpaka: The University of the West Indies, St., Augustine, Trinidad and Tobago; Özgen Köseoglu Eser: Hacettepe University Faculty of Medicine, Ankara, Turkey; Alison Maguire: University of Cambridge, Cambridge, United Kingdom; David Aanensen: Imperial College London, United Kingdom; Leon Bentley: Wellcome Sanger Institute, Cambridge, United Kingdom; Jyothish N Nair Thulasee Bhai: Wellcome Sanger Institute, Cambridge, United Kingdom; Rafal Mostowy: Imperial College London, United Kingdom; John A Lees: Imperial College London, London, United Kingdom; Keith P Klugman: Bill & Melinda Gates Foundation, Seattle, United States; Paulina Hawkins: Rollins School of Public Health, Emory University, Atlanta, United States; David Cleary: University of Southampton, Southampton, United Kingdom

## Competing interests

Nicholas J Croucher: has consulted for Antigen Discovery Inc Has received an investigator-initiated award from GlaxoSmithKline. The other authors declare that no competing interests exist.

## Funding

| Funder | Grant reference number | Author |
| --- | --- | --- |
| Wellcome Trust | 102169/Z/13/Z | Joshua C D'Aeth |
| Medical Research Council | MR/R015600/1 | Nicholas J Croucher<br>Joshua Charles D'Aeth |
| Department for International Development | MR/T016434/1 | Nicholas J Croucher<br>Joshua Charles D'Aeth |
| Wellcome Trust | Henry Dale Fellowship 104169/Z/14/A | Nicholas J Croucher |
| Bill and Melinda Gates Foundation | OPP1034556 | Stephanie W Lo<br>Rebecca A Gladstone<br>Stephen D Bentley |
| Wellcome Trust | 098051 | Stephanie W Lo<br>Rebecca A Gladstone<br>Stephen D Bentley |
| Wellcome Trust | 106698 | Paul Turner |
| National Institutes of Health | R01 AI106786 | William P Hanage |
| Royal Society | Henry Dale Fellowship 104169/Z/14/A | Nicholas J Croucher |
| Wellcome Trust | 206194 | Stephanie W Lo |

Rebecca A Gladstone
Stephen D Bentley

Centers for Disease Control       Stephanie W Lo
and Prevention                    Rebecca A Gladstone
                                  Stephen D Bentley

The funders had no role in study design, data collection and interpretation, or the decision to submit the work for publication.

## Author contributions

Joshua C D'Aeth, Conceptualization, Data curation, Software, Formal analysis, Investigation, Visualization, Methodology, Writing - original draft, Writing - review and editing; Mark PG van der Linden, Lesley McGee, Paul Turner, Stephanie W Lo, Bernard Beall, Data curation, Formal analysis, Writing - review and editing; Herminia de Lencastre, Jae-Hoon Song, Rebecca A Gladstone, Raquel Sá-Leão, Kwan Soo Ko, Data curation, Formal analysis; William P Hanage, Conceptualization, Data curation; Robert F Breiman, Conceptualization, Data curation, Project administration; Stephen D Bentley, Conceptualization, Data curation, Supervision, Writing - review and editing; Nicholas J Croucher, Conceptualization, Data curation, Formal analysis, Supervision, Investigation, Writing - original draft, Project administration, Writing - review and editing; The GPS Consortium, Data curation, Conceptualization, Resources, Formal analysis

## Author ORCIDs

Joshua C D'Aeth https://orcid.org/0000-0002-9636-9886
Paul Turner http://orcid.org/0000-0002-1013-7815
Kwan Soo Ko http://orcid.org/0000-0002-0978-1937
Nicholas J Croucher https://orcid.org/0000-0001-6303-8768

## Decision letter and Author response

Decision letter https://doi.org/10.7554/eLife.67113.sa1
Author response https://doi.org/10.7554/eLife.67113.sa2

## Additional files

### Supplementary files
• Transparent reporting form

### Data availability

All Sequencing data comes from publically available previously published datasets. All sequences used and their accession codes are available in the supporting Figure 1 source data table. All figure source data has been deposited at Figshare, https://doi.org/10.6084/m9.figshare.c.5306462.v1.

The following dataset was generated:

| Author(s) | Year | Dataset title | Dataset URL | Database and Identifier |
|---|---|---|---|---|
| D'Aeth JC | 2021 | The role of interspecies recombinations in the evolution of antibiotic resistant pneumococci | https://doi.org/10.6084/m9.figshare.c.5306462.v1 | figshare, 10.6084/m9.figshare.c.5306462.v1 |

The following previously published datasets were used:

| Author(s) | Year | Dataset title | Dataset URL | Database and Identifier |
|---|---|---|---|---|
| Wellcome Sanger Institute | 2010 | Streptococcus pneumoniae global lineages | https://www.ebi.ac.uk/ena/browser/view/PRJEB2255 | ENA, PRJEB2255 |
| Bentley SD | 2019 | Global Pneumococcal Sequencing project | https://www.pneumo-gen.net/gps/index.html | ENA, PRJEB3084 |

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
