## [Decision Letter]

**Acceptance summary:**

This paper reports a thorough, sophisticated and rigorous analysis of the evolutionary history of *Streptococcus pneumoniae* genomes with particular focus on mechanisms affecting the evolution of antibiotic resistance and immune evasion.

**Decision letter after peer review:**

Congratulations, we are pleased to inform you that your article, "The role of interspecies recombinations in the evolution of antibiotic resistant pneumococci", has been accepted for publication in *eLife*. Your article has been reviewed by 2 peer reviewers, and the evaluation has been overseen by a Reviewing Editor and a Senior Editor.

However, before final acceptance, it is really important that you pay attention to the comments of referee 1, that I fully support. All three referees agree on the care, the rigour and the sophistication of your analysis. However, in among all the detail – and there is a lot of detail – the message gets buried. We strongly urge you to take a step back, decide on the main message, and make this explicit in the abstract and use it, additionally, to focus the discussion. In the absence of such focus your super paper risks being read solely by those that care about evolution of *S. pnuemoniae*. Our belief in committing to publish in *eLife* is that your paper stands to interest a much broader audience.

*Reviewer #1:*

The authors have succeeded in a state of the art micro-dissection of evolution of two important lineages of S. pneumoniae. It is particularly interesting to see how transposable elements have been inserted and deleted.

The authors have been less successful in distilling the data towards salient observations or finding novel take-homes.

I am generally sceptical of temporal estimates of effective population size of particular lineages, especially because they can easily be influenced by sampling as well as by natural selection. The authors show that the populations are in fact very geographically stratified, so which geographical regions are included or excluded will inevitably alter the results considerably. The bacteria also live in neighbouring upsampled countries (and regions) and can hop back and forth. The evidence that antibiotic prescription has driven changes in effective population size seems particularly stretched and unconvincing.

The abstract also does not make it very clear what the analysis of the two lineages, which is the meat of the manuscript, has taught us. The final quantity given is about antibiotic spread across the species, suggesting that there is not really a scientific line of reasoning towards conclusions in the mind of the authors.

The paragraphs in the discussion are very long. This makes it hard to discern what the line of argument and points of emphasis are, if any.

I am very much in favour of this kind of careful work but do not think there is a well made argument for publication in a broad readership journal at this point and do not immediately have an idea where one will come from.

*Reviewer #2:*

By comprehensive analyses of large collections of genome sequences of Streptococcus pneumoniae isolates of the two clinically important lineages PMEN3 and PMEN9 that have spread worldwide, the authors attempt to reconstruct the evolutionary histories of these lineages and their antibiotic resistance. Using sophisticated modelling and comparisons to national antibiotic consumption data from European countries the authors demonstrate interesting differences in the evolutionary history of the two lineages. Differences include differences in the time-span of their evolution and the number of capsular switches. The study is unique in being based on large sets of representative data and comprehensive state-of-the-art analyses, and the careful conclusions are well supported. Although some of the findings are expected as they confirm previous reports, the paper provides novel information about the time-span of the evolution and the variability of evolutionary speed and genetic mechanisms within the species S. pneumoniae and conclusive evidence for previously reported assumptions.

It is obvious that the manuscript is carefully executed and describes all parts of the analyses and considerations in an exemplary way. I found a few minor points that need attention:

Lines 22-24: "under selection by community antibiotic consumption, which is driven by common noninvasive pneumococcal diseases, such as otitis media". The sentence seems to suggest that the relevant selection pressure is restricted to the treatment of infections caused by S. pneumoniae. This is of course not the case.

Line 272: It is not clear to me what "a single 491 isolate ST156 clade" means.

---

## [Author Response]

Before final acceptance, it is really important that you pay attention to the comments of referee 1, that I fully support. All three referees agree on the care, the rigour and the sophistication of your analysis. However, in among all the detail -- and there is a lot of detail – the message gets buried. We strongly urge you to take a step back, decide on the main message, and make this explicit in the abstract and use it, additionally, to focus the discussion.

We agree with the reviewing editor that the Abstract and Discussion of the paper tried to cover too many points in detail, rather than focusing on the main message of the paper. We have addressed this in the revision through comprehensively editing both to reframe the main results of the paper in terms of the main conclusion, that many resistances in the pathogenic pneumococcus have been recently acquired from related commensal species. In the Abstract, we note such transfers were consistently observed across both the PMEN3 and PMEN9 lineages, despite their otherwise divergent epidemiology, and the species-wide GPS dataset. In the shortened Discussion, we were able to synthesise the implications of the detected interspecies transformation events for the bacterial species concept, and the selection pressure imposed by antibiotic consumption. The overall paper greatly benefits from these improvements, and we thank the reviewers and editors for their recommendations.

In the absence of such focus your super paper risks being read solely by those that care about evolution of S. pnuemoniae. Our belief in committing to publish in eLife is that your paper stands to interest a much broader audience.Reviewer #1:The authors have succeeded in a state of the art micro-dissection of evolution of two important lineages of S. pneumoniae. It is particularly interesting to see how transposable elements have been inserted and deleted.The authors have been less successful in distilling the data towards salient observations or finding novel take-homes.

We appreciate this is a long manuscript that describes complex datasets in some detail, and therefore it can be difficult to focus on a single compelling conclusion. As mentioned above, we have narrowed the focus of the Abstract and Discussion to focus more on the novel inferences regarding the acquisition of mobile elements containing antibiotic resistance genes from commensal species through recent homologous recombination. As noted in the text, while individual genes and gene segments have been known to transfer between species via transformation, the movement of these larger elements between species has much wider implications for the movement of chromosomal loci in bacterial communities. We hope this leads to renewed attention into less well-studied commensal species that are often the source of these resistance genes. Therefore we believe this discovery will be of interest to the community, as repeated interspecies transformation was one of the few commonalities in the emergence of different antibiotic-resistant *S. pneumoniae* genotypes, which exhibited otherwise surprisingly divergent patterns of evolution and transmission.

I am generally skeptical of temporal estimates of effective population size of particular lineages, especially because they can easily be influenced by sampling as well as by natural selection. The authors show that the populations are in fact very geographically stratified, so which geographical regions are included or excluded will inevitably alter the results considerably. The bacteria also live in neighbouring upsampled countries (and regions) and can hop back and forth. The evidence that antibiotic prescription has driven changes in effective population size seems particularly stretched and unconvincing.

We agree with the reviewer that systematic sampling is key for this type of phylodynamic analysis to produce reliable results. As such, we only conducted this type of analysis on a single clade, across both PMEN lineages, where we were confident the data were appropriate. All isolates used in this analysis were assembled by the German national public health laboratory (Nationales Referenzzentrum für Streptokokken), which systematically collects pneumococci from cases of invasive disease across the country. Therefore, we believe this to be a reasonably-consistently sampled population, minimising any sampling bias in this analysis. Additionally, based on our global sampling, this densely-sampled clade was highly localised to Germany, minimising the “hop[ping] back and forth” between countries that the reviewer correctly suggests would weaken the link between our sampling and national prescribing policies. Additionally, regarding “which geographical regions are included or excluded”: the geographic area sampled for disease cases matches the area at which antibiotic prescribing data were aggregated, and therefore the comparison between the pharmaceutical and genomic data occurs at the same geographic scale.

Furthermore, rather than being an entirely *post hoc* analysis, this represents a statistical test of a modified version of the hypothesis originally published by Reinert et al. (2002, J. Antimicrob. Chemother. 49(1):61-68). Our revised manuscript includes an improved phylodynamic analysis, which shows that the results are robust to longer MCMCs, which still converge on a similar result, and identify a significant correlation of growth rate with antibiotic consumption whether we analyse macrolide consumption, or the ratio of macrolide and β-lactam consumption. We hope these provide more convincing evidence for the role of antibiotic consumption in the growth of this lineage.

The abstract also does not make it very clear what the analysis of the two lineages, which is the meat of the manuscript, has taught us. The final quantity given is about antibiotic spread across the species, suggesting that there is not really a scientific line of reasoning towards conclusions in the mind of the authors.

We appreciate the Abstract tried to list too many observations from the study, rather than focussing on the main conclusion of the work. As mentioned above in reply to the reviewing editor’s comments, more focus is now placed on the interspecies homologous recombinations enabling the spread of resistances carried on mobile genetic elements from commensals to pathogenic *S. pneumoniae*. We hope the revised version more accurately conveys the primary conclusions from this broad analysis.

The paragraphs in the discussion are very long. This makes it hard to discern what the line of argument and points of emphasis are, if any.

We agree with the reviewer that the Discussion covered many points raised by the analysis, at the expense of clarity. The broad scope of the results, covering vaccine evasion, resistance gain of function and the epidemiology of *S. pneumoniae* contributed to this. We have now focused the Discussion, with the main through line being the characteristics of the interspecies recombination events and what they mean with regards to questions around the nature of bacterial species.

I am very much in favour of this kind of careful work but do not think there is a well made argument for publication in a broad readership journal at this point and do not immediately have an idea where one will come from.

We thank the reviewer for their comments about the nature of the work. As described above, we have now more strongly focused parts of the work on a central message, regarding how recombination between species drives the spread of resistance between commensals and pathogens. We believe this is fundamental to two of the topics of the Special Issue (“Antibiotic and antimicrobial resistance evolution” and “The evolutionary impacts of medicine”). These data relate to questions regarding the nature of bacterial species, and how transformation affects bacterial population structure (Fraser et al. 2007 Science 315:476-480; Sheppard et al. 2008 Science 320:237-239; Hanage et al. 2005 BMC Biol. 3:6). This has long been regarded as one of the most interesting topics in bacterial evolution, and we think this epidemiological work’s relevance to such a fundamental genetic process means it will appeal to those with either a biological or clinical background.

Reviewer #2:By comprehensive analyses of large collections of genome sequences of Streptococcus pneumoniae isolates of the two clinically important lineages PMEN3 and PMEN9 that have spread worldwide, the authors attempt to reconstruct the evolutionary histories of these lineages and their antibiotic resistance. Using sophisticated modelling and comparisons to national antibiotic consumption data from European countries the authors demonstrate interesting differences in the evolutionary history of the two lineages. Differences include differences in the time-span of their evolution and the number of capsular switches. The study is unique in being based on large sets of representative data and comprehensive state-of-the-art analyses, and the careful conclusions are well supported. Although some of the findings are expected as they confirm previous reports, the paper provides novel information about the time-span of the evolution and the variability of evolutionary speed and genetic mechanisms within the species S. pneumoniae and conclusive evidence for previously reported assumptions.It is obvious that the manuscript is carefully executed and describes all parts of the analyses and considerations in an exemplary way. I found a few minor points that need attention:Lines 22-24: "under selection by community antibiotic consumption, which is driven by common noninvasive pneumococcal diseases, such as otitis media". The sentence seems to suggest that the relevant selection pressure is restricted to the treatment of infections caused by S. pneumoniae. This is of course not the case.

We thank the reviewer for highlighting this mistake. The relevant line in the Introduction has now been corrected to make clear *S. pneumoniae* infections make up only a part of the driving force behind antibiotic consumption. The altered text states: “Furthermore this resistance is under selection by community antibiotic consumption, a substantial proportion of which is often attributable to common non-invasive pneumococcal diseases, such as otitis media”.

Line 272: It is not clear to me what "a single 491 isolate ST156 clade" means.

We thank the reviewer for highlighting this confusion. This line has been updated to state, “The PMEN3 phylogeny was dominated by the 491-isolate ST156 clade”, which we hope clarifies the intended meaning.